Article 

# Dynamic patterns of functional connectivity in the human brain underlie individual memory formation

Audrey T. Phan ®[1,2], Weizhen Xie ®[1,3], Julio I. Chapeton ®[1], Sara K. Inati ®[1] & Kareem A. Zaghloul ®[1]✉

Remembering our everyday experiences involves dynamically coordinating information distributed across different brain regions. Investigating how momentary fluctuations in connectivity in the brain are relevant for episodic memory formation, however, has been challenging. Here we leverage the high temporal precision of intracranial EEG to examine sub-second changes in functional connectivity in the human brain as 20 participants perform a paired associates verbal memory task. We first identify potential functional connections by selecting electrode pairs across the neocortex that exhibit strong correlations with a consistent time delay across random recording segments. We then find that successful memory formation during the task involves dynamic sub-second changes in functional connectivity that are specific to each word pair. These patterns of dynamic changes are reinstated when participants successfully retrieve the word pairs from memory. Therefore, our data provide direct evidence that specific patterns of dynamic changes in human brain connectivity are associated with successful memory formation.

Episodic memory formation involves integrating information about an experience's content and spatiotemporal context into a unique memory trace[1]. Each memory experience may recruit neural activity across a different set of brain regions or with different temporal dynamics. For example, reading a book at the library may involve different patterns of coordinated activity in the brain as compared with listening to music at a concert. Encoding different experiences into memory should therefore involve distinct patterns of communication between different brain regions[2–4]. As these episodic experiences unfold over time, these distinct patterns of communication should also dynamically change from moment to moment[5]. Functional connectivity between different brain regions should therefore be selectively and dynamically modulated as information about each experience is integrated to form and subsequently retrieve a cohesive memory.

Previous research has investigated large-scale connectivity in the human brain[2–4,6–10] and its relevance for memory[5,11–13], but examining

the dynamic changes in neural connectivity that may underlie memory formation and retrieval has been difficult. Studies capitalizing upon the high temporal precision of scalp EEG, combined with the antomic precision of intracranial EEG (iEEG), have started to address this question. Dynamic sub-second changes in theta and high gamma frequency connectivity have been linked to memory formation and retrieval, for example[7,8,14–16]. Such studies have provided important insights into the moment-to-moment fluctuations in connectivity across brain regions that characterize successful encoding and retrieval. Even with the most temporally precise and accurate metric of communication, it still remains unclear how to interpret the dynamic modulations that occur between brain regions during individual memory events. Reading a book may involve dynamic changes in patterns of connectivity in the brain, for example, that are entirely different from the changes that arise when listening to the concert. Because of this heterogeneity, most studies of dynamic connectivity must resort to aggregating data across multiple trials or events,

[1]Surgical Neurology Branch, NINDS, National Institutes of Health, Bethesda, MD, USA. [2]Harvard–MIT Division of Health Sciences and Technology, Cambridge, MA, USA. [3]Department of Psychology, University of Maryland, College Park, MD, USA. ✉e-mail: kareem.zaghloul@nih.gov

thereby abstracting away the moment-to-moment fluctuations that are observed and likely specific to individual memories.

Thus, an important challenge for understanding how dynamic changes in large-scale brain connectivity may contribute to episodic memory formation remains how to make sense of the disparate changes in connectivity observed during different events. How do we ascribe meaning to a transient change in connectivity between a single pair of brain regions that occurs during a single trial? One approach is to examine how these patterns of dynamic changes in connectivity may be specific to individual events, and importantly, how these patterns may be reinstated during memory retrieval. If these individual dynamic changes are meaningful, then we should observe similar specific dynamic changes as individuals retrieve the same memory. Previous studies have demonstrated that dynamic changes in the oscillatory phase in local brain regions are reinstated during successful memory retrieval[17,18]. As connectivity may be related to coordinated phase relations between brain regions, this could suggest connectivity itself between brain regions could also be dynamically modulated and reinstated during memory retrieval[19].

We explicitly investigate this question here by examining intracranial recordings in the human brain as participants being monitored for seizures performed a paired-associates memory task. We leverage the high temporal and spatial precision afforded by iEEG recordings to capture sub-second changes in connectivity across distributed cortical areas in the human brain. We first identify potential functional connections by selecting electrode pairs that exhibit strong and consistent increases in the correlation between the activity recorded from the two electrodes. We require these increases in correlated activity to occur with precise and consistent timing even when evaluated across random long-duration recording segments[20,21]. We then evaluate the moment-to-moment changes in the strength of coupling between these electrode pairs at each pair's preferred time delay over the course of the memory task[22]. We find that successful memory formation involves dynamic changes in the strength of coupling between each of these identified pairs. The patterns of dynamic changes in connectivity are specific to each word pair association being encoded and retrieved and are reinstated when participants successfully retrieve associations from memory. Moreover, the reinstatement of these patterns of connectivity is separable from the reinstatement of local neural activity assessed by changes in the patterns of spectral power. Our data therefore provide direct evidence that specific patterns of dynamic sub-second changes in functional connectivity underlie successful episodic memory formation and retrieval of individual events in the human brain.

## Results

### Sub-second changes in functional connectivity during memory encoding and retrieval

We collected iEEG data from twenty participants (8 females; 33.40 ± 2.26-years-old; mean ± SEM) as they performed a paired associates episodic memory task (Fig. 1a, b; see "Methods"). Participants studied pairs of words during encoding, and after a brief math

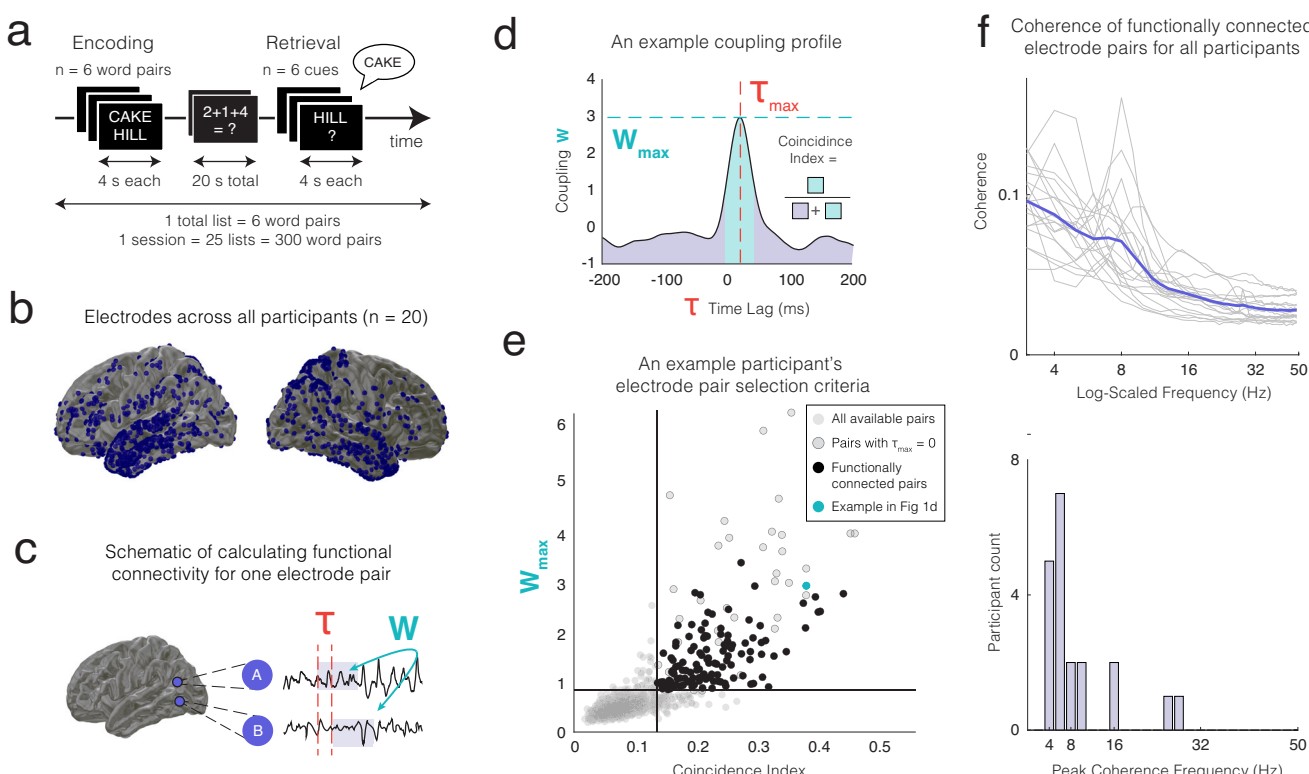

**Fig. 1 | Identifying functionally connected electrode pairs in the paired-associates task. a** The paired-associates verbal memory task. Participants are sequentially shown word pairs (encoding period), perform a math distractor task, and then cued with one of the words from each pair in random order (retrieval period). **b** Spatial coverage of intracranial electrodes implanted across participants (n = 20). **c** We compute the cross-correlation (coupling, $W$) between the iEEG time series of every electrode pair for every time lag ($\tau$). **d** The cross-correlation function across all time lags for an example pair of electrodes, computed using data from random time blocks across the entire recording session. Every electrode pair's cross-correlogram can be characterized by the maximum coupling value ($W_{max}$), preferred time lag ($\tau_{max}$), and coincidence index (CI) that captures the sharpness of the dominant peak. **e** For each participant, we used the maximum coupling and the CI values from all pairs of electrodes (grey dots) to identify functionally connected electrode pairs (black dots) that exceed a threshold set for each participant. **f** Top: spectral coherence of functionally connected electrode pairs for individual participants (grey lines) and averaged across participants (purple line), peak frequency across participants = 7 Hz. Bottom: histogram of peak frequencies across participants. Source data are provided as a Source Data file.

distraction period tried to verbally recall a word from a given word pair after being cued with the other word. Participants correctly recalled $31.22 \pm 4.91\%$ of the studied word pairs on average, with an average median reaction time of $2103 \pm 105$ ms (see Supplementary Table S2). We hypothesized that regions of the human cortex exhibit specific patterns of dynamic fluctuations in functional connectivity as participants encode word pairs into memory, and that these patterns of dynamic connectivity are reinstated when participants successfully retrieve the word pairs from memory.

To test this hypothesis, we first needed to identify pairs of brain regions that exhibit reliable functional connectivity between them, considering that many pairs of brain regions might not be functionally connected at all. We used a previously established windowed-scaled cross-correlation measure[23] to capture the functional connectivity of broadband iEEG signals between brain regions over long and randomly selected recording periods (Fig. 1c and Supplementary Fig. S1; see "Methods"). Our rationale for this approach was that if two brain regions are functionally connected, then the coupling between the regions should occur with a relatively stable time lag[20–22]. This should manifest as a clear peak in the electrode pair's cross-correlation function (see "Methods") at the pair's preferred time delay, $\tau_{max}$, with a large maximum coupling value, $W_{max}$, averaged over random recording time periods (Fig. 1d).

To select electrode pairs that exhibit strong time-locked correlations, we used a thresholding method based on the distributions of maximum coupling values and peak sharpness (coincidence index (CI)) values across pairs coupling profiles[20–22] (see Fig. 1d, e; Methods). To eliminate spurious coupling due to volume conduction, we removed all connections where the maximal coupling value, $W_{max}$, occurred with zero time delay, $\tau_{max} = 0$. In this manner, we identified an average of $371.49 \pm 33.93$ strongly connected electrode pairs per recording session across participants. Consistent with previous studies[20,22], this accounted for $11.60\% \pm 0.62\%$ of possible connections between all recorded electrodes (see Supplementary Table S3). To ensure that the identified pairs reflect stable connections, for each participant we only retained electrodes that met thresholding procedures across multiple recording sessions, which were recorded hours or days apart ($24.15 \pm 5.5$ h, mean $\pm$ SEM, Supplementary Table S3). This reduced the average number of connected electrode pairs to $184.60 \pm 26.23$ per participant, accounting for $6.35\% \pm 0.65\%$ of the possible connections between all electrodes on average. Spectral coherence analysis suggests that the average coherence frequency among connected electrode pairs reaches its peak around 6–10 Hz across participants (Fig. 1f and Supplementary Figs. S2 and S3a–c for the spatial distances between electrodes of selected pairs and their distributions of $W_{max}$ and $\tau_{max}$.)

Although we identified these functional connections based on long recording periods without time-locking to any specific task events, it is unlikely that each connected electrode pair exhibits a constant level of coupling between them during each encoding and retrieval event. Instead, such coupling should be dynamic. To examine this, we calculated the coupling strength, $W$, for each identified electrode pair at its preferred time lag, $\tau_{max}$, in 1000-ms sliding time windows (80% overlap) across the encoding and retrieval periods of the memory task (Fig. 2a; see "Methods"). We normalized these $W$ time series to measure changes in coupling from the baseline period ($W_Z$). In individual participants, we observed that some electrode pairs exhibit dynamic increases in coupling compared to baseline periods ($-1000$ ms to $-500$ ms relative to response vocalization) following stimulus onset during the encoding period, while other pairs exhibit dynamic decreases relative to baseline periods (Fig. 2b; see "Methods"). Electrode pairs demonstrating overall increases or decreases in average coupling strength during encoding appear to also show respective increases or decreases during successful retrieval. Across participants, we found that electrode pairs demonstrating increases or

decreases in coupling are drawn from overlapping spatial distributions across the cortex (Fig. 2c, d and Supplementary Table S4).

The heterogeneity of dynamic changes in coupling strength compared to baseline across electrode pairs suggests that successful memory formation may involve increased functional connectivity between some brain regions and decreased connectivity between others. To investigate the overall changes in functional connectivity that may be relevant for memory, we computed the magnitude of these changes, $|W_Z|$, in each electrode pair averaged across trials (Fig. 2e and Supplementary Fig. S4 for data from individual participants). Across participants, the magnitude of changes in coupling strength increases from baseline during the encoding period and peaks around $-1000$ ms to $-500$ ms relative to participants' response vocalization, which is referred to as retrieval time = 0 s. The magnitude of this change is significantly greater 400 ms to 3200 ms after stimulus onset for correct trials as compared with incorrect trials (cluster-based $p_{corrected} < 0.05$; cluster mean difference between correct and incorrect trials: $t(19) = 3.18$, $p = 0.0049$, Cohen's $d = 0.71$, 95% confidence interval (CI) [0.21, 1.20]). The magnitude of increase or decrease in coupling strength, $|W_Z|$, does not vary based on the distance between the two electrodes in each pair (Supplementary Fig. S5; see "Methods"). During retrieval, for both correct and incorrect trials, we observed a similar increase in the magnitude of changes in coupling strength before a participant vocally retrieved the associated word, although this increase was not significantly greater in the correct relative to incorrect trials (Fig. 2e). Together, these data demonstrate that the patterns of coupling between brain regions exhibit dynamic changes during the paired associates memory task, with greater changes on average in coupling strength during the encoding period of correct trials.

## Patterns of dynamic functional connectivity during memory encoding are reinstated during successful memory retrieval

To examine whether the changes in coupling strength between electrode pairs during encoding are reinstated during successful memory retrieval, we constructed feature vectors containing the normalized coupling strength values, $W_Z$, of each functionally connected pair at its preferred time delay for every time point during the encoding and retrieval periods of every word pair. For each participant, we then computed the cosine similarity between these feature vectors for every combination of encoding and retrieval time points (Fig. 3a; see "Methods"). This generates a similarity profile capturing the extent to which the patterns of dynamic coupling are reinstated at each time point when a participant encodes and retrieves a given word pair. We averaged these similarity profiles separately across correct and incorrect trials for each participant (Fig. 3b and see Supplementary Fig. S6 for pass and intrusion trials, see Supplementary Fig. S7 for trials split by response time). Across participants, there was significantly greater reinstatement of these patterns of dynamic coupling in correct trials as compared with incorrect trials, peaking around $1360 \pm 200$ ms after study onset and $-550 \pm 170$ ms relative to vocalization (cluster-based $p_{corrected} < 0.05$; temporal region of interest (tROI); Fig. 3c and Supplementary Fig. S8). Within this identified tROI, the mean encoding-retrieval similarity averaged across participants is significantly greater during correct compared to incorrect trials ($t(19) = 3.60$; $p = 0.0019$; Cohen's $d = 1.03$, 95% CI [0.48, 1.47]; Fig. 3c).

The greater reinstatement of dynamic coupling we observe during correct encoding and retrieval could reflect unique patterns of coupling that are specific to the encoding and retrieval of individual word pairs or general encoding and retrieval mechanisms that are deployed across trials. To test whether these patterns of reinstatement are specific to individual word pairs, we compared reinstatement observed during correct trials with that computed when we shuffled the correct trial retrieval labels[24] (see Supplementary Fig. S9 for reinstatement of first-presented retrieval items). If the reinstatement of dynamic functional connectivity contains pair-specific information beyond generic

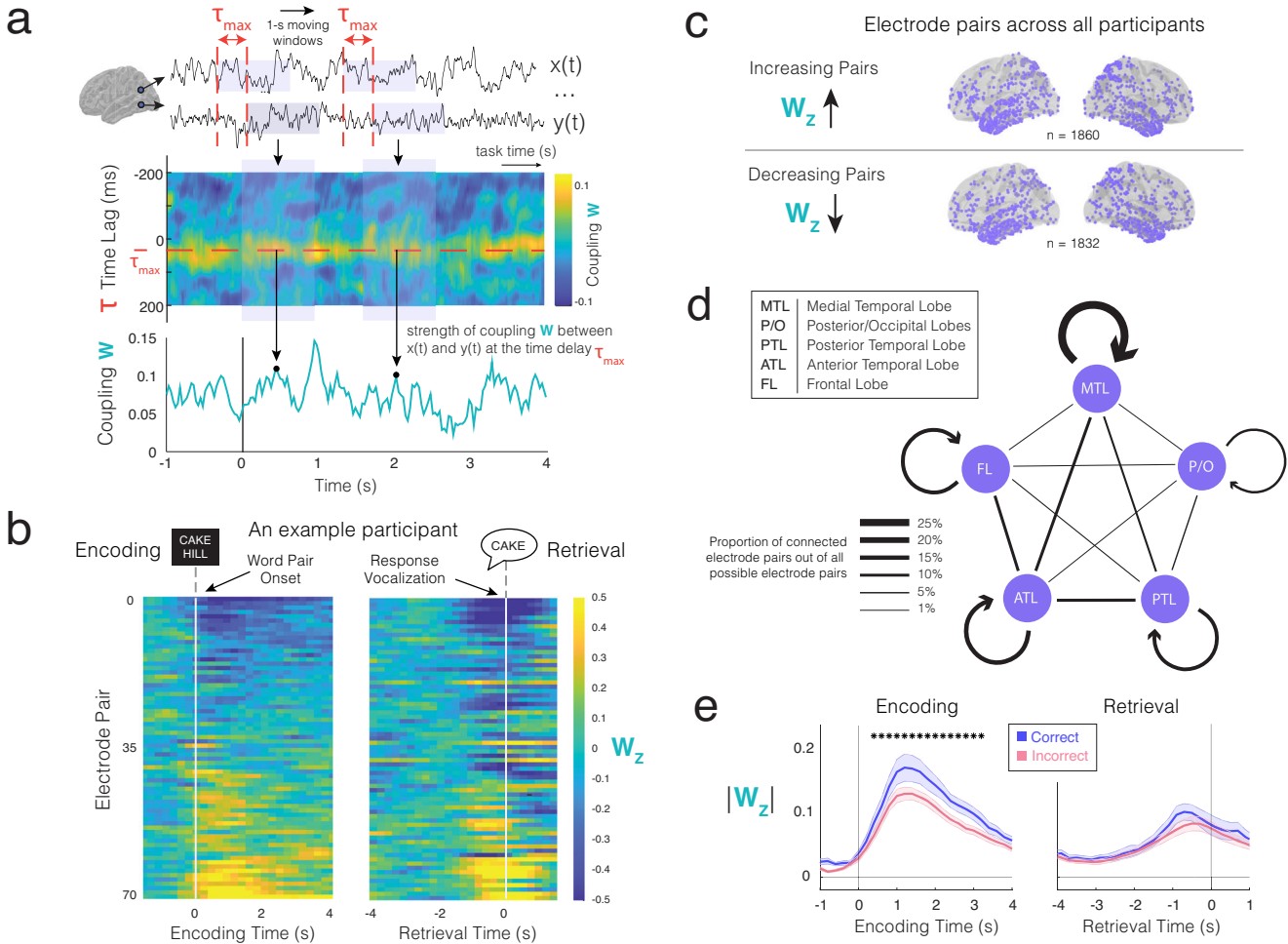

**Fig. 2 | Sub-second changes in functional connectivity during memory encoding and retrieval. a** For each electrode pair, we calculate the moving-window correlation at that pair's preferred time lag, $\tau_{max}$, across encoding and retrieval. **b** Changes in coupling strength relative to baseline ($W_Z$) (see "Methods") for all pairs in an example participant, across encoding and retrieval. Pairs show dynamic increases or decreases in $W_Z$ following encoding stimulus onset, with similar dynamics during retrieval. Pairs are sorted by average $W_Z$ during encoding. **c** Pairs with average increases in coupling during memory formation (top) and those with average decreases (bottom). **d** Schematic of functionally connected electrode pairs

within and across brain regions. Line thickness represents the proportion of identified electrode pairs out of all possible pairs for the involved brain region(s). **e** Across participants ($n = 20$), memory encoding and retrieval are associated with significant increases in the magnitude of changes in coupling strength ($p < 0.05$, cluster-based permutation test). During encoding (but not retrieval), this change is significantly greater for correct compared to incorrect trials; $p_{corrected} < 0.05$, cluster-based permutation test, two-sided). Error bands represent SEM across participants. Source data are provided as a Source Data file.

encoding and retrieval processes, then the similarity observed between encoding and retrieval during the original correct trials should be greater than that observed using shuffled correct retrieval trial labels. Supporting this prediction, we find that the average encoding-retrieval similarity in the tROI is significantly greater in the original correct trials as compared with that in the correct trials with shuffled labels ($t(19) = 3.55$; $p = 0.0021$, Cohen's $d = 0.79$, 95% CI [0.28, 1.29]; Fig. 3d). Furthermore, when we used the labels of adjacent correct retrieval trials, we also found that the encoding-retrieval similarity of the original correct trials remains significantly greater than that of the adjacent correct trials within the tROI ($t(19) = 3.91$; $p = 0.00093$; Cohen's $d = 0.87$, 95% CI [0.35, 1.39]; Fig. 3d). These results suggest that patterns of dynamic coupling during memory encoding are reinstated during successful retrieval and that this reinstatement is specific to the individual word pair encoded and retrieved from memory.

### Reinstatement of dynamic functional connectivity and spectral power are separable

Although we examined dynamic coupling between pairs of functionally connected electrodes, we were interested in confirming that such

changes in coupling are not simply a consequence of, nor redundant with, the changes in spectral power. It is possible that the reinstatement of the changes in coupling between electrode pairs could arise due to the reinstatement of local spectral power. For example, the timing of coupling reinstatement is consistent with previous research investigating the reinstatement of spectral power during the paired associates memory task[24–26]. To exclude this possibility, we directly compared the reinstatement of coupling with the reinstatement of spectral power within each participant. We computed the reinstatement of spectral power across individual electrodes between encoding and retrieval, similar to prior studies[24]. For each participant during every time point, we constructed feature vectors containing the oscillatory power from every electrode in five frequency bands (theta, 3.5–8 Hz; alpha, 8–12 Hz; beta, 13–25 Hz; low gamma, 30–58 Hz; high gamma, 62–100 Hz; see "Methods"). We then computed the cosine similarity between the feature vectors at every pair of encoding and retrieval time points to generate a profile of reinstatement for each trial. Similar to our previous analysis, we compared these reinstatement profiles between correct and incorrect trials across participants and identified a tROI in which there is significantly greater

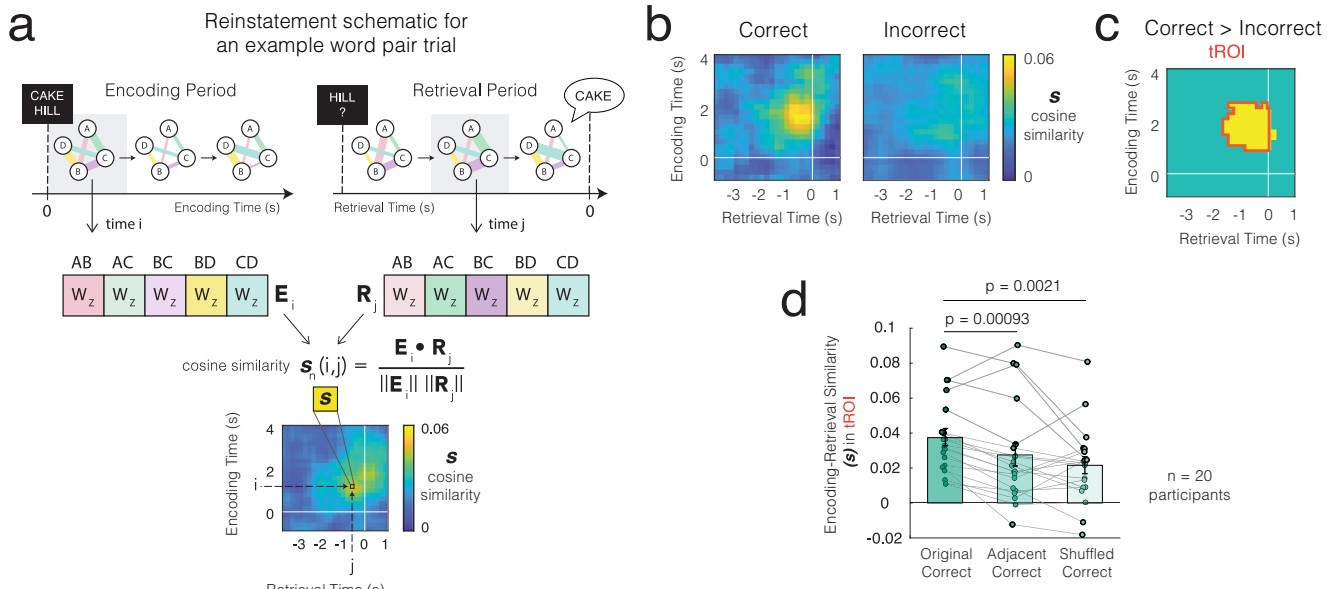

**Fig. 3 | Patterns of dynamic functional connectivity during memory encoding are reinstated during successful memory retrieval. a** For every time point during encoding and retrieval periods, we construct feature vectors of normalized coupling values ($W$) of every functionally connected electrode pair. For every pair of encoding and retrieval time windows ($i$, $j$), we calculate the cosine similarity, $s$, between their feature vectors ($E_i$ and $R_j$) to generate a similarity map for all pairs of encoding and retrieval time windows. **b** Average reinstatement map showing encoding-retrieval similarity across all participants ($n = 20$) during correct and incorrect trials, time-locked to stimulus onset for encoding, and vocalization for retrieval. **c** We define the tROI as encoding-retrieval time pairs before vocalization that exhibit significantly greater similarity incorrect compared with incorrect trials across participants. tROI outlined in *red*; cluster-based permutation test ($p < 0.05$). Within the tROI, the mean encoding-retrieval similarity averaged across participants is significantly greater during correct compared to incorrect trials ($p = 0.0019$, two-sided *t*-test). **d** Mean similarity of patterns of dynamic coupling between encoding and retrieval in the tROI across participants is greater for the original correct retrieval trial labels as compared with shuffled correct retrieval trial labels, and greater for the original correct retrieval trial labels as compared with adjacent correct retrieval trial labels. Error bars represent SEM, with individual participants' results shown as dots and connected lines. $N = 20$ participants. Two-tailed uncorrected *p*-values were calculated based on a paired-sample *t*-test. Source data are provided as a Source Data file.

reinstatement of spectral power in the correct trials relative to incorrect trials prior to vocalization (Supplementary Fig. S10b). This tROI peaks around $1050 \pm 340$ ms after study onset and $-1000 \pm 330$ ms relative to vocalization, replicating the previous findings[24].

We then computed the relative contribution of each individual electrode to the reinstatement of both coupling and spectral power using a leave-one-out approach[26–28]. We systematically excluded one electrode or electrode pair, for spectral power and functional connectivity respectively, and recomputed the difference in reinstatement in the tROI between the original and the leave-one-out conditions, thereby providing an estimate of the relative contribution of that electrode or electrode pair to overall reinstatement in the tROI (see "Methods" for details; Supplementary Fig. S11). This procedure generates for each electrode a measure capturing its contribution to the reinstatement of spectral power. As the reinstatement of coupling involves pairs of electrodes, we average the contributions of all electrode pairs that contain a given electrode to estimate the contribution of that electrode to the reinstatement of coupling. We did not observe specific regional localization of brain areas that contribute more towards the reinstatement of spectral power or dynamic connectivity (see Supplementary Fig. S11 for spatial electrode distributions, Supplementary Fig. S3c, d for distributions of $W_{max}$).

To investigate the relation between the reinstatement of dynamic coupling and the reinstatement of spectral power at the electrode level, we first correlated each electrode's relative contributions to the reinstatement of coupling and spectral power within each participant (Supplementary Fig. S12). Across participants, we did not find a systematic correlation pattern between the contributions to the reinstatement of coupling and spectral power across electrodes (average Fisher's transformed $r = 0.021$, $t(19) = 0.83$, $p = 0.42$, Cohen's $d = 0.19$,

95% CI [−0.26, 0.63]). Even after correcting for the attenuation in correlations that may arise due to noisy estimates of each electrode's contribution to reinstatement[29], the correlation remains statistically not significant (average Fisher's transformed $r = 0.051$, $t(19) = 0.67$, $p = 0.51$, Cohen's $d = 0.15$, 95% CI [−0.29, 0.59]).

Next, to further reveal how the reinstatement of coupling is separable from the reinstatement of local changes in spectral power, we also divided each participants' electrodes into those that provide a higher vs. lower contribution to the reinstatement of coupling based on a median split. We then separately recomputed the reinstatement of coupling and spectral power using these two different sets of electrodes (Fig. 4a). By construction, electrodes with a higher contribution to the reinstatement of coupling exhibit a significantly stronger reinstatement of coupling relative to those with a lower contribution within our tROI identified when originally computing the reinstatement of coupling ($t(19) = 6.41$, $p = 0.0000038$, Cohen's $d = 1.43$, 95% CI [0.80, 2.06]; see tROI in Fig. 3b). If the reinstatement of coupling and the reinstatement of power are closely related, we would expect to see a similar difference between these groups of electrodes when computing the reinstatement of spectral power. However, splitting electrodes in the same manner does not lead to a similar change in the reinstatement of power within the tROI ($t(19) = 2.06$, $p = 0.053$, Cohen's $d = 0.46$, 95% CI [−0.006, 0.92]; Fig. 4b), see tROI of spectral power analysis in Supplementary Fig. S10b). Instead, splitting the electrodes based on their relative contributions to the reinstatement of coupling has led to a significantly greater change in the reinstatement of coupling as compared with that of spectral power (interaction between electrode group and reinstatement measure: $F(1,19) = 37.52$, $p = 0.0000069$, $\eta_p^2 = 0.66$) (Fig. 4b). Both sets of electrodes, with high and low contributions to the reinstatement of coupling respectively, demonstrate reliable reinstatement of spectral

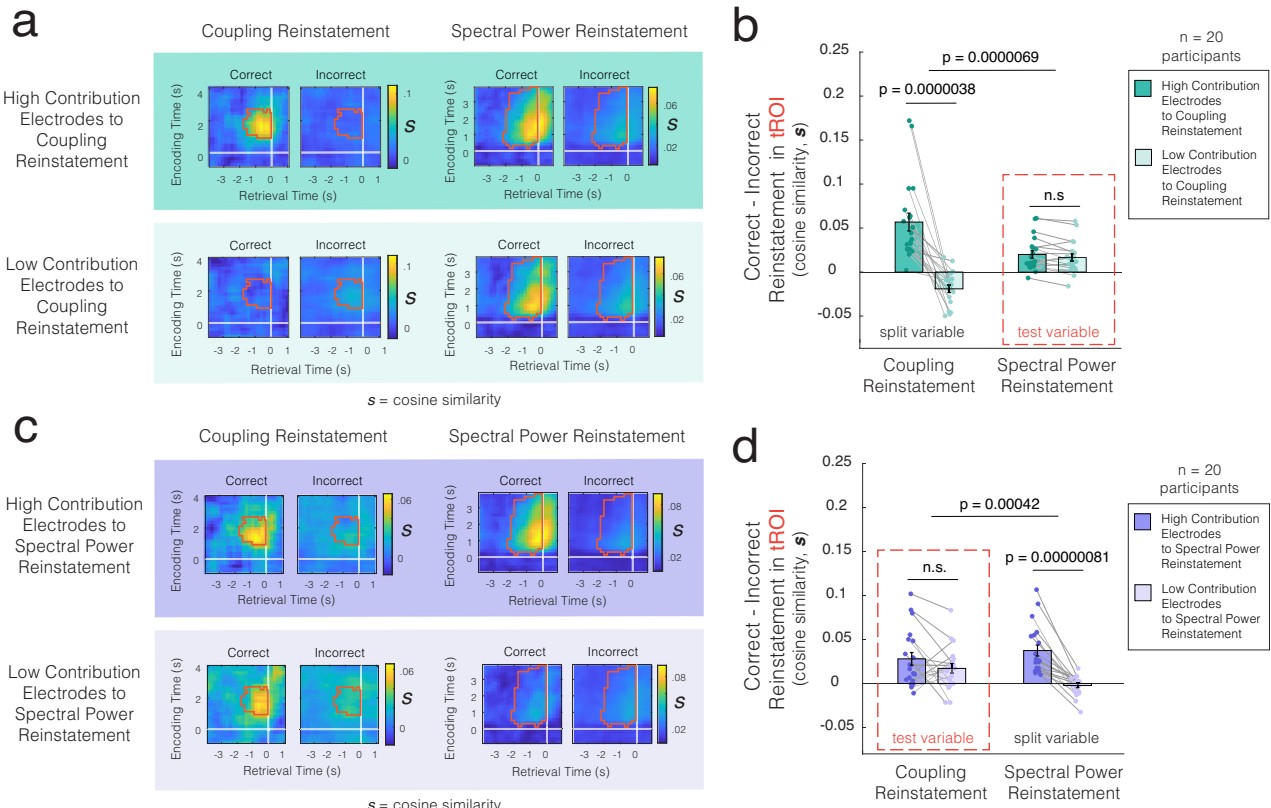

**Fig. 4 | Reinstatement of dynamic functional connectivity and spectral power are separable. a** Reinstatement of coupling (left) and spectral power (right) after dividing electrodes into high (top) and low (bottom) contributors to coupling reinstatement (tROIs outlined in *red*). **b** High-contributing electrodes show significantly greater coupling reinstatement within the tROI, while this split does not significantly affect spectral power reinstatement ($n = 20$, two-sided $t$-tests, no correction). The interaction between the electrode group and the reinstatement measure is significant (two-way ANOVA). **c** Reinstatement of coupling (left) and

spectral power (right) after dividing electrodes into high (top) and low (bottom) contributors to spectral power reinstatement (tROIs outlined in red). **d** High-contributing electrodes exhibit significantly greater spectral power reinstatement, but this split does not significantly affect coupling reinstatement ($n = 20$, two-sided $t$-tests, no correction). The interaction between the electrode group and the reinstatement measure is significant (two-way ANOVA). Individual data are shown as dots. Error bars represent SEM across participants. *n.s.* not significant. Source data are provided as a Source Data file.

power within the tROI (electrodes showing high vs low contribution to coupling reinstatement: $t(19) = 4.70$, $p = 0.00015$, Cohen's $d = 1.05$, 95% CI [0.49, 1.59] vs $t(19) = 4.11$, $p = 0.00060$, Cohen's $d = 0.91$, 95% CI [0.38, 1.44], respectively; Fig. 4a). These data demonstrate that splitting the electrodes based on their relative contributions to the reinstatement of coupling has a minimal effect on the reinstatement of power.

We similarly split each participant's electrodes into those that provide higher vs lower contributions to the reinstatement of spectral power (Fig. 4c). By construction again, we find that electrodes with a higher contribution to the reinstatement of power exhibit a significantly greater reinstatement of spectral power as compared with those showing a lower contribution within the tROI identified when originally computing the reinstatement of spectral power ($t(19) = 7.18$, $p = 0.00000081$, Cohen's $d = 1.61$, 95% CI [0.93, 2.27]; see tROI in Supplementary Fig. S10b). If the reinstatement of coupling and the reinstatement of spectral power are redundant, we would expect to see a similar difference when computing the reinstatement of coupling. However, this split does not lead to similar changes in the reinstatement of coupling ($t(19) = 1.45$, $p = 0.16$, Cohen's $d = 0.32$, 95% CI [−0.13, 0.77]; Fig. 4d, see tROI of coupling analysis in Fig. 3c). Instead, splitting the electrodes based on their relative contributions to the reinstatement of spectral power only leads to a significantly greater change in the reinstatement of spectral power but not in the

reinstatement of coupling (interaction between electrode group and reinstatement measure: $F(1,19) = 18.21$, $p = 0.00042$, $\eta_p^2 = 0.49$; Fig. 4d). Regardless of whether a set of electrodes contributes more or less to the reinstatement of spectral power, both sets of electrodes show reliable reinstatement of coupling within the tROI (electrodes showing higher vs lower contribution to power reinstatement: $t(19) = 3.90$, $p = 0.00096$, Cohen's $d = 0.87$, 95% CI [0.35, 1.38] vs $t(19) = 3.09$, $p = 0.0060$, Cohen's $d = 0.69$, 95% CI [0.19, 1.17] respectively; see Fig. 4c). These data demonstrate that splitting the electrodes based on their relative contributions to the reinstatement of spectral power has a minimal effect on the reinstatement of coupling. Together, these dissociable patterns suggest that the reinstatement of patterns of dynamic changes in functional connectivity and the reinstatement of patterns of local spectral power are separable and are not redundant.

## Discussion

Our data demonstrate that functional connections between brain regions are dynamically modulated to support remembering specific events. In our current task, every memory event related to a word pair involves creating and remembering a unique association. Representing and encoding these unique associations may therefore involve distinct patterns of functional connectivity between brain regions. Supporting this hypothesis, we find that patterns of dynamic

coupling between different brain regions at encoding are reinstated during successful retrieval and specific to each individual word pair being remembered. The reinstatement of these patterns of functional connectivity is separable from the reinstatement of local spectral power[24,25]. Together, these findings demonstrate that the brain's ability to encode and retrieve information is not just restricted to local patterns of neural activation, but also involves moment-to-moment adjustments of specific patterns of connectivity across brain regions.

Although it has been explicitly shown that the brain is a networked structure[2–4,30–33], characterizing how dynamic fluctuations in connectivity between brain regions give rise to successful remembering has been overshadowed by research focused on the functional localization of memory[34–38]. Because the events that we experience change from moment to moment, however, and because the specific brain regions involved in processing those experiences will also change from moment to moment, rapidly forming and retrieving memories of those experiences should also involve dynamic changes in connectivity between different brain regions[19]. Studies that rely upon neuroimaging cannot capture these sub-second changes in connectivity because of its poor temporal resolution, although recent studies have demonstrated changes in task-locked functional connectivity that may have significant behavioural relevance[39,6,40,9,11–13,41].

Studies deploying scalp and iEEG can address this question given their temporal resolution and ability to capture neural network structures using measures such as coherence, cross-correlation, Granger causality, or phase locking[7,8,42–44]. These studies have provided important insights regarding the role of dynamic connectivity in memory formation[19]. These approaches have often aggregated data across multiple trials to provide an overall average measure of how connectivity dynamically changes across different memory states or over long recording durations[14–16]. Connectivity between different brain regions, however, should be more heterogeneous, changing from moment to moment in different ways between different brain regions since information represented in the brain at any one moment is itself dynamic and constantly in flux[19,45]. It has been challenging to draw inferences regarding the rapid changes in neural connectivity that may be specific to individual memory events.

To address this challenge, we examined connectivity during both encoding and retrieval of the same memory. If a pair of electrodes exhibit increased connectivity when encoding a word pair association into memory and exhibits a similar increase when retrieving that same word pair association from memory, then that increase is likely to be functionally meaningful. Furthermore, because representing and encoding each unique episode into memory should involve coordinating unique information across multiple different brain regions, then the pattern of dynamic changes across all pairs of electrodes that occur when encoding each memory should be reinstated when retrieving that same information from memory. Our data support these predictions. Successful memory retrieval involves significantly greater reinstatement of the dynamic patterns of connectivity as compared to unsuccessful memory retrieval. Moreover, the similarity between encoding and retrieval within correct trials is greater than that when examining shuffled or adjacent correct trials, suggesting that the observed reinstatement of connectivity does not simply reflect a generic mechanism common to all correct retrieval trials. Our findings instead provide direct evidence that successful retrieval of each individual memory involves reinstatement of a specific pattern of dynamic functional connectivity.

Our results build upon substantial evidence that has suggested that successful memory retrieval involves the reinstatement of local neural activity that was present when memory was first encoded[24–26,46,47]. If specific patterns of dynamic connectivity across brain regions are also relevant when encoding a particular event into memory, similar patterns of connectivity should also be dynamically reactivated when that information is retrieved from memory[1]. Conceptually, reinstating dynamic patterns of cortical connectivity may help efficiently recapitulate the initial neural representation of a past event and thereby increase the likelihood of successful remembering as compared to relying on local spectral power alone. Indeed, we find that the reinstatement of local power is complementary to yet separable from the reinstatement of patterns of connectivity. The patterns of dynamic connectivity reflect information regarding the specific association being remembered, complementing the information captured by patterns of local oscillatory power.

Before investigating how patterns of connectivity between brain regions dynamically fluctuate from moment to moment, we needed to identify pairs of electrodes with reliable functional connections between them. We reasoned that the only fluctuations in connectivity that should be meaningfully examined are those that occur through stable functional connections[20]. We implemented a strict requirement for identifying these potential functional connections based on the premise that if two brain regions are connected, then any coupling between them should occur with a relatively constrained time delay[20,22,48–50]. By way of a loose analogy, we are inferring the presence of a phone line that connects two households by tracking the activity of the residents of the two homes over a long time period and finding that their activity is correlated more often than expected by chance and with a relatively consistent time delay. By examining correlated activity over time blocks that are randomly sampled throughout the recording session, we were able to identify electrode pairs that satisfy this requirement. The identified pairs comprise a sparse subset of all possible connections, consistent with empirical results regarding functional brain connectivity[20,22]. Critically, the strength of these correlations fluctuates throughout different periods of the task. We therefore focused our subsequent analysis on the time-varying version of this metric to capture the dynamics of connectivity across each identified electrode pair specifically at their preferred time delay. To extend the analogy, we are asking if and precisely when the residents of the two households are actually talking from moment to moment, although we cannot specifically infer what they are actually saying.

Our study builds upon recent work investigating the network mechanisms underlying episodic memory[5,7,8,11–18]. While previous studies on brain activity during episodic memory have focused on more localized brain areas[34–38], we adopted a broader approach to examine all possible cortical electrode pairs recorded in each participant. As episodic memory involves integrating different aspects of an experience represented in different brain areas, it is likely that connectivity is distributed across brain networks[13]. In line with this hypothesis, we found that different patterns of dynamic connectivity spanning a diverse set of cortical brain regions were involved when participants formed and retrieved individual word pairs. We did not find any specific hubs contributing more to the reinstatement of dynamic connectivity. This suggests that probing and possibly manipulating functional connectivity in the brain during memory formation would thus likely require focusing on specific connections for specific memory events.

It has been challenging to reveal the precise temporal dynamics of connectivity occurring within the sub-second timescales that characterize the rapid formation and retrieval of individual episodic memories. Our data provide evidence that dynamic changes in specific patterns of connectivity may be functionally relevant for successful memory formation. Despite the broad distribution of functional connectivity that we observe, the patterns of dynamic changes in connectivity between brain regions are specific to the memory being formed, consistent with the fact that every event we experience involves unique information. Our findings therefore

highlight the importance of rapid neural reinstatement during episodic memory across multiple spatial scales of the human brain, from single units[46,47], to local brain regions[24–26], to large-scale cortical networks.

## Methods

### Participants

Twenty participants (8 females; 33.40 ± 2.30 years old; Wechsler Intelligence Quotient [IQ]: 88.17 ± 2.67, all >70; see Supplementary Table S1) with drug-resistant epilepsy underwent a surgical procedure in which platinum recording contacts were implanted on the cortical surface as well as within the brain parenchyma. In each case, the clinical team determined the placement of the electrode contacts to localize epileptogenic regions. During seizure-free recording sessions, participants completed the English version of a paired associates memory task. Some of the data from this sample have been published in previous studies examining neural reinstatement underlying successful memory formation and retrieval at the level of local neural circuits[24,25,46,51,52], but not at the level of large-scale cortical network as in the current study. Based on these prior studies, we used the following inclusion criteria for the analysis of neural data: (1) participants should have no prior resection of brain regions; (2) they should have completed at least TEN accurate trials of the memory task across experimental sessions to ensure reasonable signal stability for neural reinstatement analysis[25]; and (3) the recorded iEEG data should not contain excessive environmental noise or motor artefacts (see Preprocessing for details). Data were collected at the Clinical Centre at the National Institutes of Health (NIH; Bethesda, MD). The Institutional Review Board (IRB) of the National Institutes of Health and the National Institute of Neurological Disorders and Stroke approved the research protocol, and informed consent was obtained from the participants and their guardians. No sex or gender-based analyses were performed as they were not relevant to this study's research questions.

### Paired associates memory task

Each participant performed a paired associates memory task in which we presented lists of study and test word pairs. During the study period, participants are sequentially shown a list of word pairs (encoding period) and instructed to remember the novel associations between each pair of words. Later during testing, they are cued with one word from each pair selected at random (retrieval period), and are instructed to vocalize the associated word into a microphone. A 'trial' during this task refers to the encoding and retrieval period of a one-word pair.

A single experimental session for each participant consists of 25 lists, where each list contains 6 pairs of common nouns shown on the center of a laptop screen. The number of pairs in a list is kept constant for each participant. Words are chosen at random and without replacement from a pool of 300 high-frequency nouns and are presented sequentially and appear in capital letters at the centre of the screen. In order to ensure that memory formation and retrieval are not directly adjacent in the task, study word pairs are separated from their corresponding retrieval cue by a minimum lag of two study or test items. During the study period (encoding), each word pair is preceded by an orientation stimulus (+) that appears on the screen for 250 ms followed by a blank inter-stimulus interval (ISI) between 500 ms and 750 ms. Word pairs are then presented stacked in the centre of the screen for 4000 ms followed by a blank ISI of 1000 ms. Following the presentation of the list of word pairs, participants complete an arithmetic distractor task of the form A + B + C = ? for approximately 20,000 ms.

During the subsequent test period (retrieval), one word is randomly chosen from each of the presented pairs and presented in random order, and the participant is asked to recall the other word from the pair by vocalizing a response. Each cue word is preceded by an orientation stimulus (a row of question marks,"????") that appears on the screen for 250–300 ms followed by a blank ISI of 500–750 ms. Then, cue words are presented on the screen for 4000 ms followed by a blank ISI of 1000 ms. Participants can vocalize their responses at any time during the retrieval period after cue presentation. We filtered out any trial where the participant took longer than 4000 ms to respond after the word presentation. In our analyses, participants' time of response is marked as retrieval time = 0 s. We manually coded the task responses and designated each response as correct, intrusion, or pass. For correct responses, we ignored utterances such as "umm…" prior to response and manually assigned the response vocalization time to when the participant began vocalizing the correct word. A response is considered a pass when no vocalization was made or when the participant vocalized the word 'pass'. During trials where no vocalization was present, we assigned a response time by randomly drawing from the distribution of correct response times during that experimental session in order to constrain the range of possible response times and align all trials by time for subsequent analyses. We defined all intrusion and pass trials as incorrect trials. A single experimental session contained 60–150 total word pairs and 10–25 lists within the 1-h recording session. We included at most 2 unique sessions from each participant to minimize repeated exposure to the same words[25] (see Supplementary Table S2 for trial counts per participant).

### Intracranial recordings and pre-processing

We recorded iEEG signals from subdural and stereotactic EEG (sEEG) electrode contacts (PMT Corporation, Chanhassen, MN) using a Nihon Kohden (Tokyo, Japan) or Blackrock Microsystems (Salt Lake City, UT) data acquisition system. For this study, we examined data only from the subdural electrode contacts in each participant. Subdural electrode contacts were arranged in grid or strip configuration with an inter-contact spacing of 5–10 mm and covered a wide range of cortical areas, with the most consistent coverage across participants overlying the anterior temporal lobe (Fig. 1b). We localized these electrode contacts by co-registering the post-op CTs with the pre-op MRIs of each individual participant's brain using a previously established method[53].

Depending on the amplifier and the discretion of the clinical team, iEEG signals were sampled at 1000 or 2000 Hz. The recorded raw iEEG signals used for analyses were resampled at 1000 Hz. We rejected electrode contacts from our analysis that exhibited abnormal signal amplitude or large line noise, including those with an average amplitude or variance of voltage trace greater than three standard deviations above the mean across all recorded electrode contacts[21,54]. We then removed slow fluctuations in the iEEG time series using a local detrending procedure, removed line noise at 60 Hz and 120 Hz using a regression-based approach[55], and low-passed the data at 150 Hz to remove higher order line harmonics as well as high-frequency noise. We re-referenced the resulting signals using bipolar referencing based on the immediate adjacent electrode contacts to mitigate any effects of volume conduction or any biases introduced by the system hardware reference. The location of bipolar-referenced signals was defined by the midpoint between adjacent electrode contacts. Henceforth, we refer to these bipolar-referenced signals as electrodes. In total, following these preprocessing steps, we retained 1536 bipolar-referenced electrodes for our analysis (range: 31–132; average: 76.80 ± 5.25 per participant; also see Supplementary Table S2).

We extracted epochs of the clean iEEG signals for each encoding and retrieval trial. For each encoding trial, we extracted an epoch of 2000 ms before to 5000 ms following the onset of the study word pair. For each retrieval trial, we extracted an epoch 5000 ms before to 2000 ms following either the time point of vocalization or the randomly assigned response time for that trial if the participant did not respond. We included a 1000 ms buffer at the beginning and end of each encoding and retrieval epoch. For analyses of spectral power, we

quantified spectral power by convolving iEEG signals with complex-valued Morlet wavelets (40 logarithmically spaced values from 3 Hz to 150 Hz, wavelet number 6). We then squared and log-transformed the continuous-time wavelet transform to generate a continuous measure of instantaneous power. After removing the 1000-ms buffered data at the beginning and end of each epoch, we extracted the baseline-normalized power values for each frequency and each electrode separately using the across-trial mean and standard deviation of the pre-trial baseline power from −700 ms to −500 ms before stimulus onset[56].

## Functional connectivity

We used a previously established method to investigate the directed functional connectivity between every recorded electrode pair in our data[20–22]. We quantified time-locked correlations in broadband activity (1–150 Hz) between every electrode pair with the rationale that if information is communicated across brain regions via a stable pathway, then their activity should be strongly correlated over a broad frequency range with a consistent time delay[20,21]. To estimate these functional connections while ensuring that participants were awake and behaving, we randomly sampled 20 blocks of 30-s data within each recording session (about 1/6 of the total recording time) without time-locking to individual task events (Supplementary Figs. S1 and S13 for task-only sampling). Although some blocks include time periods during the task, previous work has shown that these types of network connections and their time delays are stable across minutes, hours, and days, across different times and tasks[20,22]. We used a previously established windowed-scaled cross-correlation measure[23] to capture the functional relationship of broadband iEEG signals between brain regions over long and randomly selected recording periods (see Supplementary Fig. S1). Specifically, for each electrode pair in each block of randomly selected iEEG data, we extracted 1–150 Hz broadband activity and computed the absolute value of the correlation between the respective time series data in overlapping 1000 ms time windows (80% overlap, windows shifted by 200 ms) with a varying temporal offset (t) ranging from −200 ms to 200 ms in 1 ms steps. The resulting values averaged across all moving time windows of the randomly selected data provide robust estimates of the absolute cross-correlation function between two-time series, $\overline{R(\tau)}$. We defined the coupling function, $W$(t), between every electrode pair and for every block of random data, as the cross-correlation function $\overline{R(\tau)}$ normalized by the mean and standard deviation of $R$ overall $\tau$:

$$W(\tau) = \frac{\overline{R(\tau)} - \mu_R}{\sigma_R} \qquad (1)$$

We then averaged the estimates of the coupling function $W$(τ) across the randomly selected data blocks. Electrodes that exhibit consistent delays when computing the cross-correlation of their time series will exhibit an average cross-correlation function with a robust peak. Hence, the maximum value of the averaged coupling function, $W_{max}$, provides a measure of how consistently correlations between the two signals are time-locked at their associated time delay ($\tau_{max}$) relative to all other delays (Fig. 1d). Using these $W_{max}$ and $\tau_{max}$ values for every electrode pair, we also calculated the widely used CI to characterize the sharpness of the peak in the coupling profile[22,57–59]. We operationalized the CI as the normalized full width at half the maximum area under the curve of the dominant peak (Fig. 1d).

To examine dynamic communication between brain regions, we identified functionally connected pairs of electrodes by examining the coupling function for each electrode pair. We identified the effective functional connections that exhibit robust cross-correlations at a preferred and a relatively fixed time lag based on the following thresholding heuristic. We fit the distributions of $W_{max}$ and CI from all available electrode pairs with a mixture of two Gaussians. We defined a

knee point that occurs between the two means and used that value as the threshold for connected vs unconnected pairs[21] (see Fig. 1e for an example). We also removed all electrode pairs where the $W_{max}$ occurred with zero time delay, $\tau_{max} = 0$ since these functional connections can arise spuriously[7,20,60], for example, due to volume conduction[60] (see Supplementary Fig. S14 for results after inclusion of these electrode pairs). To capture electrode pairs that are reliably functionally connected across time, we further selected electrode pairs only if they met thresholding criteria across all sessions for each participant, which were often recorded hours or days apart (Supplementary Table S3). In this manner, we identified functionally connected electrode pairs that show a high coupling value, $W_{max}$, with a consistent and preferred time delay, $\tau_{max}$. To understand which frequencies may be involved in functional connectivity, we computed the spectral coherence for each identified pair over the task period (Fig. 1f). We also identified the broad anatomic regions involved in each electrode pair to understand how these functionally connected electrode pairs are distributed across brain regions (Fig. 2d, Supplementary Table S4, and Supplementary Fig. S11).

After identifying functionally connected pairs of electrodes, we next investigated how their coupling strength ($W$) changes as a function of time during the paired-associates verbal memory task. Importantly, the identification of functionally connected electrode pairs is based on randomly sampled epochs throughout the recording session, while we examine how the strengths of these connections dynamically fluctuate over time only during the task period. The analysis of time-varying changes in connectivity should therefore not be circular. The randomly sampled time blocks include data outside of the task itself, such as when the participant is conversing with staff or taking a break. Furthermore, previous evidence has demonstrated that these identified functional connections exhibit stable time delays and connectivity across a variety of task conditions[20]. Thus, conceptually, the approach we adopt here first identifies electrode pairs agnostic to their role in the memory task, and then asks how these pairs change their connections during memory formation and retrieval.

We calculated the sliding window correlation of the broadband 1–150 Hz filtered iEEG traces between two connected electrodes at their preferred time lag $\tau_{max}$ using 1000 ms moving windows (80% overlap) in each encoding and retrieval trial (see Fig. 2a for an example). We then normalized these time series of coupling strength, $W_Z$, for each electrode pair using the across-trial mean and standard deviation of the pre-trial baseline coupling for that electrode pair, computed −1000 ms to −500 ms before stimulus onset (baseline period). For visualization in individual participants (see Fig. 2b for an example participant), we sorted pairs in ascending order using $W_Z$ values averaged across the entire encoding period (0–4000 ms following stimulus onset) over all correct encoding trials. We retained the same order for sorting the pairs when visualizing the dynamic changes in coupling strength for these electrode pairs during retrieval.

To identify which electrode pairs showed increases or decreases in coupling, we calculated the pair's average change in coupling during encoding (0–4000 ms following stimulus onset). If $W_Z$ is positive or negative following stimulus onset, we categorized that electrode pair as increasing or decreasing in connectivity, respectively. To examine the magnitude of the changes in coupling strength as a function of time, irrespective of whether the change in coupling involves an increase or a decrease, we computed the absolute value of the average normalized time series of coupling strength for each electrode pair, separately for all correct and incorrect trials (Supplementary Fig. S4 and Fig. 2e).

## Neural reinstatement

To quantify the extent to which neural signals are reinstated during successful memory retrieval, we separately calculated encoding-

retrieval similarity for dynamic coupling across functionally connected electrode pairs and for spectral power across electrodes. For every time point during encoding ($i$) and every time point during retrieval ($j$), we constructed a feature vector for encoding ($\mathbf{E}_i$) and one for retrieval ($\mathbf{R}_j$) based on the neural data aggregated across electrodes.

To examine reinstatement of patterns of dynamic coupling across electrode pairs, we created feature vectors that contain the normalized coupling strength values, $W_Z$, for all functionally connected electrode pairs ($p = 1...P$) and for every encoding ($i$) and retrieval time point ($j$),

$$\mathbf{E}_i = [W_{z1}(i)...W_{zp}(i)...W_{zP}(i)] \tag{2}$$

$$\mathbf{R}_j = [W_{z1}(j)...W_{zp}(j)...W_{zP}(j)] \tag{3}$$

where $W_{zp}(i)$ is the normalized instantaneous coupling strength $W_Z$ at encoding time point $i$ for each electrode pair $p$ using that electrode pair's preferred time delay as identified through the cross-correlation procedure above. In the same manner, $W_{zp}(j)$ is the normalized instantaneous coupling strength $W_Z$ at retrieval time point $j$ for each electrode pair $p$ using that electrode pair's preferred time delay.

To examine the reinstatement of spectral power, we replicated a previous study[24] and created feature vectors for encoding and retrieval that contain power information from all recorded electrode locations and five frequency bands.

(theta, 3.5–8 Hz; alpha, 8–12 Hz; beta, 13–25 Hz; low gamma, 30–58 Hz; and high gamma, 62–100 Hz),

$$\mathbf{E}_i = [z_{1,1}(i)...z_{1,F}(i)...z_{K,F}(i)] \tag{4}$$

$$\mathbf{R}_j = [z_{1,1}(j)...z_{1,F}(j)...z_{K,F}(j)] \tag{5}$$

where $z_{k,f}(i)$ is the normalized power of electrode $k = 1...K$ at frequency band $f = 1...F$ at time point $i$. Hence, each feature vector contained $K \times F$ features, which represent the distributed spectral power across all electrodes and across five frequency bands at a single moment in time.

The reinstatement of a given study-and-test trial at each encoding and retrieval time point can be formalized as the cosine similarity, $S(i,j)$, between these encoding and retrieval feature vectors for every combination of encoding time $i$ and retrieval time $j$,

$$S(i,j) = \frac{\mathbf{E}_i \cdot \mathbf{R}_j}{||\mathbf{E}_i|| \cdot ||\mathbf{R}_j||} \tag{6}$$

In this manner, we generated a precise temporal map of neural reinstatement between the encoding and retrieval periods based on spectral power or coupling for every trial. Then, we separately averaged temporal maps over correct and incorrect trials in every participant to generate a single map of the reinstatement profile for each participant. We defined a tROI separately for the reinstatement of coupling (Supplementary Fig. S10a) and the reinstatement of spectral power (Supplementary Fig. S10b) by selecting all encoding-retrieval time pairs (excluding retrieval times after vocalization) that showed significantly greater encoding-retrieval similarity between correct and incorrect trials.

### Spectral power vs coupling reinstatement
To determine the extent to which the reinstatement of coupling and the reinstatement of power may be related, we quantified the relative contribution of each electrode to these measures of reinstatement

using a leave-one-out procedure[26–28]. We systematically excluded each electrode or electrode pair and recomputed the differences in mean reinstatement in the tROI between correct and incorrect trials across participants. Excluding an electrode or electrode pair, $el$, in this manner generates a measure of the relative contribution ($C$) of that element to overall differences in the reinstatement we observe:

$$C = \frac{S_{all} - S_{all-el}}{|S_{all}|} \tag{7}$$

where $S_{all}$ is the mean difference in reinstatement in the tROI between correct and incorrect trials across participants when using all electrodes, and where $S_{all} - e_l$ is this difference after excluding that electrode or electrode pair. When computing the contribution to the reinstatement of spectral power, we excluded the measures of spectral power from the five frequency bands used to compute power reinstatement for each electrode. For coupling, as each electrode can be part of different connected electrode pairs, we first excluded one electrode pair at a time to calculate the contribution of each electrode pair to the reinstatement of coupling. We then took the average of these contribution values for all electrode pairs involving a given electrode to generate the contribution of that individual electrode to the reinstatement of coupling.

We then normalized these contributions separately for power and coupling by scaling these values within each participant from 0 and 1,

$$C_{norm} = \frac{C - C_{min}}{C_{max} - C_{min}} \tag{8}$$

which generates a normalized estimate of each electrode's relative contribution to the reinstatement of spectral power and coupling, $Cpow_e$ and $Cconn_e$, respectively. Electrodes with normalized contributions close to 0 provide the least contribution to reinstatement, while those with normalized contributions close to 1 provide the most. For each participant, we calculated the correlation between the electrodes' normalized contributions to power reinstatement and their normalized contributions to coupling reinstatement (Fig. 4).

To examine the correlation between these contribution scores across participants, we corrected for the attenuation in correlation potentially due to the noise in estimating these values[29,61]. Specifically, we estimated the reliability, or internal consistency, of $Cpow_e$ and $Cconn_e$ separately within each participant by recalculating these estimates based on resampled data across 100 iterations. In each iteration, we resampled the correct and incorrect trials twice with replacement and estimated the correlations between $Cpow_e$ and $Cconn_e$ in these two random samples of the data. We took the average of these correlation estimates across iterations as the mean reliability estimates, $r_{xx}$ or $r_{yy}$, where $x$ and $y$ represent the two variables of interest, $Cpow_e$ and $Cconn_e$, respectively. The attenuation-corrected correlation ($r'_{x y}$) between the two variables of interest is therefore estimated as:

$$r'_{xy} = \frac{r_{xy}}{\sqrt{r_{xx} \times r_{yy}}} \tag{9}$$

To more directly visualize the relationship between the reinstatement of coupling and spectral power, we divided the electrodes into different groups based on a median split of their relative contributions to either the reinstatement of coupling or spectral power. We then examined whether groups of electrodes with higher or lower contributions to the reinstatement of coupling would impact the reinstatement of spectral power, and vice versa (Fig. 4). The rationale is that if these reinstatements are separable, grouping the set of electrodes based on the relative contribution to one reinstatement

phenomenon should not affect the other. Different from the reinstatement of spectral power in individual electrodes, the reinstatement of coupling is calculated based on pairs of electrodes. We therefore categorized electrode pairs as having higher or lower contributions only if both individual electrodes in a connected pair have higher or lower contributions, respectively, to the reinstatement of coupling.

## Statistical analysis

We used conventional statistical procedures such as paired-sample t-tests, repeated-measures ANOVA, and resampling analyses to obtain participant-level estimates of effect sizes and significance levels. To examine the reinstatement between trial types (correct–incorrect), we computed reinstatement profiles in correct and incorrect trials for each participant. We then compared these reinstatement profiles between trial types across participants (e.g., Fig. 3b, c), using a combination of a permutation test and cluster-wise correction as previously described[24,56]. We defined a significant cluster as one in which the true cluster statistic exceeds the distribution of permuted cluster statistics across 1000 iterations at the level of 0.05. We used this significant cluster to define tROIs for subsequent analyses (Supplementary Fig. S10). All p-values reported are two-tailed. We used a similar cluster-based correction procedure when examining the magnitude of the changes in coupling strength observed over time.

## Reporting summary

Further information on research design is available in the Nature Portfolio Reporting Summary linked to this article.

## Data availability

Processed data used in this study can be found at: https://research. ninds.nih.gov/zaghloul-lab/downloads. Source data are provided as a Source Data file. Source data are provided with this paper.

## Code availability

The custom code used in this study can be found at: https://research. ninds.nih.gov/zaghloul-lab/downloads.

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

## Acknowledgements

We thank Samantha Jackson, Oceane Fruchet, Molly Baumhauer, Nicholas Faturos, Danielle McAuliffe, Joshua Diamond, Kelsey Sundby, Uma Mohan, Kohleman Swift, and John Wittig Jr. for their contributions to data collection, data preprocessing, and insightful comments on the project. This work was supported by the Intramural Research Programs of the National Institute for Neurological Disorders and Stroke (ZIA-NS003144, K.A.Z). W.X. receives additional support from the NIH Pathway to Independence Award (K99NS126492, W.X.). Finally, we are deeply appreciative of all patients who have selflessly volunteered their time to participate in this study.

## Author contributions

A.T.P., W.X., J.I.C., and K.A.Z. conceptualized the study. A.T.P., W.X., J.I.C., S.K.I, and K.A.Z. developed the methodology and collected the data. A.T.P. and W.X. analysed the data. A.T.P., W.X., and K.A.Z. wrote the manuscript. K.A.Z. provided supervision.

## Competing interests

The authors declare no competing financial interests or competing non-financial interests.
