## [Peer Review File · Nature Communications]

Dynamic patterns of functional connectivity in human cortical networks are specific to individual memory formationReviewer #1 (Remarks to the Author):

The paper by presents an analysis predicated on the idea that connectivity during associative memory encoding and retrieval exhibits precise temporal variability, and that this pattern of connectivity will exhibit reinstatement during successful recollection at the time of test.

Executing this analysis/testing this hypothesis is complex, because coupling at different frequency ranges may exhibit different temporal dynamics. The authors employ what I think is a pretty novel approach for episodic memory experiments to identify maximum connectivity using task agnostic signal across the entire time series and then using connection pairs that exceed a threshold for further analysis. The writing is clear and the statistical presentation is straightforward and reasonable. The dataset is fairly unique, including a mix of electrode types. The downside to this approach is that the heterogeneity of the connections (in terms of region and frequency band) makes it difficult to draw any conclusions about the functional role of specific brain networks or oscillations that may support memory, but the insights presented by the authors are a unique perspective on episodic memory activity.

Elimination of spurious connection at lag=0 makes some sense, but not over long distances (ie, inter—hemispheric or distant intra hemispheric connections) connections are unlikely to be due to volume conduction. Although, I agree that such zero lag coupling would likely reflect response to some kind of common input, for what it's worth. Did the authors attempt some kind of distance cut off for exclusion of coupling values? This may explain partly why the temporal lobe exhibits such robust effects, given the preponderance of such electrodes in the dataset.

P Assoc in some sense creates challenges for analysis of encoding retrieval similarity because the timing of vocalization will vary across trials. Did the authors execute an analysis to examine dynamic connectivity examining responses with similar RTs, or perhaps using the RT as a regressor in their threshold-determination analysis for significant connectivity that goes into the reinstatement analysis? Or was the response restricted to a set time frame after item onset? In that case, doing a response locked time analysis wouldn't make sense I suppose.

Method seems to select for regions that exhibit elevated connectivity in response to stimulus for both rec and non rec pairs. This may omit regions that exhibit connectivity increases specifically in successful encoding. Are the patterns of connectivity (in terms of temporal dynamics and center frequency) different if the screening analysis is applied to successful encoding events only?

A related point is that the authors report a reinstatement memory effect that is certainly expected based on previous findings (sort of a sanity check). If I understand, the reinstatement vector consists of band limited cross correlation across all channels that meet some threshold for connectivity originally. But I think this threshold was defined using all time points, which means it was defined using both encoding and retrieval data, which sort of sets up the likelihood that it will show encoding/retrieval similarity in the connectivity information. In other words, electrode pairs that do not reinstate would show less connectivity across the whole time series and would be thresholded out of inclusion. I would be curious how the reinstatement vector predicts memory success when defined using encoding only. The time series includes both correct and incorrect trials, so I would suspect that the memory effect persists, but the authors should show that.

The authors compare item—item similarity vectors to adjacent items in an effort to account for drift that would exhibit item—specific reinstatement. To really conclude that there is item level information, the authors should show that a shuffle using semantic information shows greater persistence than temporal adjacency, as the item information should be more similar for semantically related items.

Finally, when interpreting recalled/non—recalled effects in paired associates, the signal will be sensitive to the order that items were presented at test. I.e., presentation of an item will improve recall for adjacent items, especially if successfully retrieved. This effect is different than the adjacency of items at encoding, which is discussed above, and can be somewhat tricky in PA. The authors should perhaps show that connectivity memory effects persist for the first—presented item at the time of test, or use a model that accounts for the order at which items were presented at test.

Reviewer #1 (Remarks on code availability):

The analyses are pretty straightforward.

Reviewer #2 (Remarks to the Author):

In their manuscript, Phan and colleagues present an analysis of intracranial EEG recorded while participants performed a paired associates cued recall task. The specific question was whether there patterns of functional connectivity related to encoding and retrieval of specific items.

While analysis of functional connectivity has greatly expanded our understanding of the neural correlates of memory encoding and retrieval, it has done so on the average level, due to either the slow time course of the data analyzed (e.g., fMRI) or the general approach of aggregating over many events. To address this shortcoming, the authors developed a novel approach that allowed for assessing the dynamic changes in functional connectivity arising from individual encoding and retrieval events. A critical aspect of their findings is that the dynamic changes in functional connectivity they observed are not due to general effective memory states, but actually track item-specific encoding and retrieval processes.

This manuscript is an important contribution to the field, demonstrated through a creative and sophisticated analysis of a neural activity collected while participants performed a canonical recall memory paradigm. The methods, while complicated, are solid and explained clearly. Below I list some minor comments that I hope assist the authors as they revise their manuscript.

- p5: It was clearly stated that no event locking was used for the initial WSCC analysis, but was it conducted with time periods that included the task? Based on Figure S1 I assume these blocks came from task periods, but can you be more specific what is meant by "random recording time periods"? Note, I see this in the methods later, but it may be good to further discuss why you picked random chunks of data. On a related note, do you get the same connectivity patterns (and time delays) if you perform this initial WSCC analysis on data from a completely different

time/task?

- p5: I realize you refer to the methods here, but might it be possible to give a one-sentence description of the "common thresholding heuristic" used to select electrode pairs? Or is the description for the rest of this paragraph summarizing that heuristic? As written it seems like there was a threshold applied and then the pairs were further reduced.

- p7: The shuffling procedure to demonstrate the item-specific reinstatement of functional connectivity patterns is extremely important and one of the most interesting results presented. Did the same hold for the spectral analysis?

- p8, l221: Is the claim that there is not a significant difference in reinstatement power between the high vs. low coupling electrodes resting on the $p=0.053$? Meaning that it's just above the standard 0.05 threshold? If so, that's a quite weak claim. If not, can you explain your test in more detail? Also, Figure S8 has the caption Figure S6, such that there are two S6 figures. That said, it does not seem like Figure S6b addresses the claim. I am, however, convinced by the entire pattern of results for the split-half analyses that the spectral and coupling reinstatement is dissociable.

- p11, l282: Drawing RTs from correct trials to fill in the "pass" trials seems like you're making up data. Why do this at all? I read awhile later in the methods that this is to specify the range for non-responses. It's likely worth mentioning this here.

- p11, l305: How do you specify the location of the bipolar referenced electrodes? Do you split the difference between the pair?

- f3: There is a subfigure e in the caption that does not exist in the figure.

Reviewer #2 (Remarks on code availability):

Although I did not download the data, I only saw a data download at that link, not a separate download for code. It could be that there is code provided with the data.

Reviewer #3 (Remarks to the Author):

In the paper titled 'Dynamic Patterns of Functional Connectivity in Human Cortical Networks Are Specific to Individual Memory Formation,' the authors explored the role of functionally connected electrode pairs in memory formation. They observed a higher connection strength between electrode pairs in correct trials compared to incorrect ones during encoding.

Additionally, they found the reinstatement of the functional connectivity pattern during the retrieval session for word-association. Furthermore, they dissociated the reinstatement of functional connectivity pattern and power pattern.

While I found the results intriguing, there are a few aspects that the authors could clarify:

1. To eliminate potential influences from vocal preparation on encoding-retrieval similarity differences between correct and incorrect trials. It might be beneficial to segregate the incorrect condition into wrongly responded trials and non-response trials.

2. Could the author describe the behavioural performance clearly?
3. Could the authors provide information on the distribution of T_{max}? Does the spatial distance between connected electrode pairs and non-connected pairs differ? How does the autocorrelation attenuate with time within each electrode? Will the auto-correlation affect the cross-correlation?
4. Typo here? 'Across participants, the magnitude of changes in coupling strength increases from baseline during the encoding period and peaks around -1000 to -500 ms before response vocalization.'
5. Exploring how the reinstatement of connectivity pattern and power pattern, when dissociated, relates differently to behavioral performance is crucial. Additionally, elucidating the spatial distributions of the involved electrodes, such as the differing brain regions, would be valuable.
6. Could the authors provide information on the distribution of functional connectivity between electrodes contributing to power pattern reinstatement?

Reviewer #3 (Remarks on code availability):

NA

April 16, 2024

Thank you for offering us the opportunity to revise our manuscript, “Dynamic patterns of functional connectivity in human cortical networks are specific to individual memory formation” for publication in Nature Communications. We thank the reviewers for their insightful comments that we feel were extremely helpful and that have significantly improved our manuscript. We have provided new analyses and substantially revised our manuscript based on their suggestions, and we feel that these revisions have strengthened our findings and conclusions.

We directly address the reviewer concerns here. The reviewer comments are reproduced in bold font, and we have addressed each point below the corresponding comment. We have provided a version of the manuscript highlighting all the relevant changes using red font as well as a clean pdf version of the revised manuscript.

Reviewer #1 (Remarks to the Author):

The paper by presents an analysis predicated on the idea that connectivity during associative memory encoding and retrieval exhibits precise temporal variability, and that this pattern of connectivity will exhibit reinstatement during successful recollection at the time of test.

Executing this analysis/testing this hypothesis is complex, because coupling at different frequency ranges may exhibit different temporal dynamics. The authors employ what I think is a pretty novel approach for episodic memory experiments to identify maximum connectivity using task agnostic signal across the entire time series and then using connection pairs that exceed a threshold for further analysis. The writing is clear and the statistical presentation is straightforward and reasonable. The dataset is fairly unique, including a mix of electrode types. The downside to this approach is that the heterogeneity of the connections (in terms of region and frequency band) makes it difficult to draw any conclusions about the functional role of specific brain networks or oscillations that may support memory, but the insights presented by the authors are a unique perspective on episodic memory activity.

We thank the Reviewer for their positive comments and appraisal of our manuscript.

Elimination of spurious connection at lag=0 makes some sense, but not over long distances (ie, inter—hemispheric or distant intra hemispheric connections) connections are unlikely to be due to volume conduction. Although, I agree that such zero lag coupling would likely reflect response to some kind of common input, for what it’s worth. Did the authors attempt some kind of distance cut off for exclusion of coupling values? This may explain partly why the temporal lobe exhibits such robust effects, given the preponderance of such electrodes in the dataset.

We thank the Reviewer for raising this point. We elected to remove all connections with zero lag since it is unclear to us what such zero-lag connectivity may represent. This could be volume conduction, as the Reviewer indicated, or it could be the result of a third common input. To err on the side of caution, we have removed these electrode pairs from our analyses. We did not, as the

Reviewer inquired, implement a distance cutoff for excluding these connections. We have now clarified this point in the revised Methods as shown below.

*“We also removed **all electrode** pairs where the W_{max} occurred with zero time delay, $\tau_{max} = 0$, since these functional connections can arise spuriously, for example due to volume conduction.”*

P Assoc in some sense creates challenges for analysis of encoding retrieval similarity because the timing of vocalization will vary across trials. Did the authors execute an analysis to examine dynamic connectivity examining responses with similar RTs, or perhaps using the RT as a regressor in their threshold-determination analysis for significant connectivity that goes into the reinstatement analysis? Or was the response restricted to a set time frame after item onset? In that case, doing a response locked time analysis wouldn't make sense I suppose.

In our version of the paired associates task, the response was restricted to a set time frame after item onset. Each participant has a maximum of 5 seconds from the onset of the word cue until the next stimulus appeared. Each word cue is displayed on the screen for 4 seconds followed by blank screen of 1 second. In practice, we excluded trials in which participants responded after 4 seconds because we did not want the change of screen to confound our analyses. We also performed a response-locked analyses, as the Reviewer has suggested. Finally, to account for trials in which no response was vocalized, we assigned the response an RT by selecting from the distribution of RTs. We have now clarified all of these points in the revised Results and Methods as shown below.

*“Each cue word is preceded by an orientation stimulus (a row of question marks, "????") that appears on the screen for 250-300 ms followed by a blank ISI of 500-750 ms. **Then**, cue words are presented on the screen for 4,000 ms followed by a blank ISI of 1,000 ms. Participants can vocalize their response any time **within** the retrieval period after cue presentation. **We filtered out any trial where the participant took longer than 4,000 ms to respond after word presentation. In our analyses, participants' time of response is marked as retrieval time = 0 s.**”*

*“Across participants, the magnitude of changes in coupling strength increases from baseline during the encoding period and peaks around -1000 to -500 ms **relative to participants' response vocalization, which is referred to as retrieval time = 0.**”*

Method seems to select for regions that exhibit elevated connectivity in response to stimulus for both rec and non rec pairs. This may omit regions that exhibit connectivity increases specifically in successful encoding. Are the patterns of connectivity (in terms of temporal dynamics and center frequency) different if the screening analysis is applied to successful encoding events only?

A related point is that the authors report a reinstatement memory effect that is certainly expected based on previous findings (sort of a sanity check). If I understand, the reinstatement vector consists of band limited cross correlation across all channels that meet some threshold for connectivity originally. But I think this threshold was defined using all time points, which means it was defined using both encoding and retrieval data, which sort of sets up the likelihood that it will show encoding/retrieval similarity in the connectivity information. In other words, electrode pairs that do not reinstate would show less connectivity across the

whole time series and would be thresholded out of inclusion. I would be curious how the reinstatement vector predicts memory success when defined using encoding only. The time series includes both correct and incorrect trials, so I would suspect that the memory effect persists, but the authors should show that.

We thank the Reviewer for raising these points, which we agree are important to clarify. To be clear, our approach for identifying electrode pairs that exhibit significant overall connectivity is to identify these pairs using data drawn from random time blocks sampled across the entire recording session. These time blocks are not locked to any individual trial, and include both task and non-task periods. Such non-task periods could include, for example, times when participants are speaking with hospital staff, taking a phone break, taking a bathroom break, etc. Our motivation here was to identify brain regions, and pairs of regions, that appear to have consistently elevated level of connectivity between them on average throughout all of these periods. The Reviewer is correct in that these time blocks may also include periods of successful encoding and retrieval. These blocks also include many other time points that may or may not be directly related to the task. We adopted this approach because previous work in our lab has demonstrated that these network connections and their time delays remain stable across minutes, hours, and even days, regardless of which specific behavior an individual is exhibiting during that time (Chapeton 2017, Xie 2023). As with our analyses here, in those previous studies we analyzed data across minutes, hours, and days, across different states including tasks, sleep and waking periods, eating and talking. In all cases, the connectivity between the identified electrode pairs remained stable. Additionally, almost all participants in our study here completed 2 separate experimental sessions of this task (17 of 20 participants). We only selected electrode pairs that exhibited significant connectivity in both sessions, often hours or days apart, ensuring that the observed connectivity in these electrode pairs was not specific to only one recording session. We now show the timing between sessions in Supplementary Table S3, and have clarified all of these points in our revised Methods as shown below.

“To capture electrode pairs that are reliably functionally connected across time, we further selected electrode pairs only if they met thresholding criteria across sessions for each participant, which were often recorded hours or days apart (24.15 +/- 5.5 hours, mean +/- SEM, Supplementary Table S3). In this manner, we identified functionally connected electrode pairs that show a high coupling value, W_{max} , with a consistent and preferred time delay, τ_{max} .”

“We quantified time-locked correlations in broadband activity (1-150 Hz) between every electrode pair with the rationale that if information is communicated across brain regions via a stable pathway, then their activity should be strongly correlated over a broad frequency range with a consistent time delay (ChapEtal17, ChapEtal19). To estimate these functional connections while ensuring that participants were awake and behaving, we randomly sampled 20 blocks of 30-second data within each recording session (about 1/6 of the total recording time) without time-locking to individual task events (Supplementary Figure S1). Although some time blocks include time periods during the task, previous work has shown that these network connections and their time delays are stable across minutes, hours, and days, across different times and tasks (Chapeton 2017).

The Reviewer, however, raises an interesting question regarding what our connectivity profile may look like if we restrict our analysis only to identify connected electrode pairs during encoding or successful encoding. We show below the coupling profile for one example electrode pair of one

participant, generated using three different time sampling methods: first, our original method shown in the paper taking time blocks across the recording session; second, taking time blocks only from encoding trials; and third, taking time blocks only during successful encoding trials. Across the three different methods, there is a very reliable and stable coupling profile for this pair, suggesting that the profiles we have identified are consistent whether we use time blocks distributed throughout the entire task or only times during the encoding trials.

We also believe it is important to clarify that our method does not necessarily select for pairs that show only elevated connectivity in response to the stimulus. Instead, as demonstrated in Figure 2b, around half of the pairs we have identified with overall strong connectivity in fact exhibit a decrease in connectivity in response to stimulus during the actual memory task itself. Additionally, as we show in Figure 2e, memory encoding and retrieval is associated with an increase in the magnitude of coupling strength across pairs, which includes both increases and decreases in coupling. This means that even if a pair exhibits an overall increase in coupling throughout the experimental session, that pair does not necessarily increase during the encoding or retrieval.

Finally, it is also important to clarify that even if a pair is identified with increases in coupling during both encoding and retrieval task time periods, these overall increases in connectivity are not relevant for our analysis. Instead, once we select a connected pair, we are interested in how that pair's coupling changes over encoding versus retrieval at its specific maximum time lag. As shown above in our illustrated example, the max time lag for a given pair remains stable regardless whether we sample our time blocks for cross correlation analysis from across the entire recording session, encoding time period, or successful encoding time periods. However, from moment to moment, that connectivity or coupling between that electrode pair will exhibit dynamic changes, and our primary interest here is in how those dynamic changes may be reinstated during successful memory retrieval and specific to the item or association being encoded into and retrieved from memory.

The authors compare item—item similarity vectors to adjacent items in an effort to account for drift that would exhibit item—specific reinstatement. To really conclude that there is item level information, the authors should show that a shuffle using semantic information shows greater persistence than temporal adjacency, as the item information should be more similar for semantically related items.

The Reviewer raises a good point. Our stimuli, however, are presented as word pairs, and our task tests the memory of the association between words within a pair. While each word does have semantic meaning, it is difficult to assign semantic meaning to a word pair and compare semantic meanings across word pairs. We have clarified this point in revised Results as shown below.

*“The greater reinstatement of dynamic coupling we observe during correct encoding and retrieval could reflect unique patterns of coupling that are specific to the encoding and retrieval of individual memory items **word pairs** or general encoding and retrieval mechanisms that are deployed across trials. To test whether these patterns of reinstatement are **item-specific specific to individual word pairs**, we compared reinstatement observed during correct trials with that computed when we shuffled the correct trial encoding labels. If the reinstatement of dynamic functional connectivity contains **item-specific pair-specific** information beyond generic encoding and retrieval processes, then the similarity observed between encoding and retrieval during the original correct trials should be greater than that observed using shuffled correct trial labels.”*

In addition, to address this point, we have now also introduced a new analysis in which we examined reinstatement during trials in which participants make intrusions. In an empirical analysis, we show that when individuals make intrusions, the intruded word is more semantically similar to the cue word than the expected word. In those cases, the patterns of dynamic connectivity also appear to exhibit some reinstatement. Thus, these new analyses provide some evidence supporting the Reviewer's point that patterns of connectivity shows some similarity for semantically related items. We present this analysis in a new supplementary figure, Supplementary Figure S6.

Finally, when interpreting recalled/non—recalled effects in paired associates, the signal will be sensitive to the order that items were presented at test. I.e, presentation of an item will improve recall for adjacent items , especially if successfully retrieved. This effect is different than the adjacency of items at encoding, which is discussed above, and can be somewhat tricky in PA. The authors should perhaps show that connectivity memory effects persist for the first—presented item at the time of test, or use a model that accounts for the order at

which items were presented at test.

We thank the Reviewer for raising this interesting point. We agree that there could be an interesting relation between adjacent items at the time of retrieval.

We apologize for the confusion, but it appears we had a typo in our initial submission. When performing the analyses investigating item-specificity, we performed a shuffling analysis in which we shuffled or swapped trials. In our submitted manuscript, we had claimed that we shuffled or swapped encoding trials and compared those shuffled trials to the same true retrieval trials. However, in reality, we actually performed the reverse – we shuffled or swapped retrieval trials and compared them to the true encoding trials. We believe this is the exact analysis the Reviewer has requested, and we have now corrected this error in our revised Results and in the corrected Figure 3d as shown below.

*“To test whether these patterns of reinstatement are item-specific, we compared reinstatement observed during correct trials with that computed when we shuffled the correct trial **encoding retrieval** labels. If the reinstatement of dynamic functional connectivity contains item-specific information beyond generic encoding and retrieval processes, then the similarity observed between encoding and retrieval during the original correct trials should be greater than that observed using shuffled correct trial labels. Supporting this prediction, we find that the average encoding-retrieval similarity in the tROI is significantly greater in the original correct trials as compared with that in the correct trials with shuffled labels ($t(19) = 3.55$; $p = 0.0021$; Cohen’s $d = 0.79$, Figure 3d). Furthermore, when we used the labels of adjacent **correct retrieval** trials, we also found that the encoding-retrieval similarity of the original correct trials remains significantly greater than that of the adjacent correct trials within the tROI ($t(19) = 3.91$; $p = 0.00093$; Cohen’s $d = 0.87$; Figure 3d).”*

*“d) Mean similarity of patterns of dynamic coupling between encoding and retrieval in the tROI across participants is greater for the original correct **retrieval** trial labels as compared with shuffled correct trial labels and with adjacent correct **retrieval** trial labels. Individual participant data shown as dots”*

From this analysis, we can see that there is still some reinstatement for encoding trials with adjacent correct retrieval trials, but this reinstatement is significantly less than the reinstatement observed during true correct retrieval trials.

In addition, to further address this point, we have now conducted a new analysis to test whether reinstatement persists when examining only the first presented items in each list. Given the limited amount of first trials within a list for each participant and the even smaller subset of those first trials that are correct versus incorrect, we only included participants with at least 10 correct and at least 10 incorrect first-in-list trials. This resulted in 12 participants which we included for our analysis. Within this smaller cohort, we were still able to replicate our main findings, demonstrating that reinstatement is significantly stronger for correct compared to incorrect trials. We have now included this new analysis as a new Supplementary Figure S8, also shown below.

Supplementary Figure 8

Supplementary Figure X. Reinstatement analysis for trials that are first in their list during retrieval period. We conducted an analysis to test whether the reinstatement effect persists for only first-presented items in each list at retrieval during the task. We only included participants with at least 10 correct and 10 incorrect first-in-list trials, which was $n = 12$ participants. Here we show reinstatement maps averaged over these 12 participants. Across these participants, there is significantly greater reinstatement of these patterns of dynamic coupling in correct first-in-list trials as compared with incorrect first-in-list trials, (cluster-based $p_{corrected} < 0.05$; temporal region of interest, tROI; Figure 3c; Supplementary Figure S7).

Reviewer #1 (Remarks on code availability):
The analyses are pretty straightforward.

We agree with the Reviewer that the analyses are straightforward. Nonetheless, our download link includes both code and data.

Reviewer #2 (Remarks to the Author):

In their manuscript, Phan and colleagues present an analysis of intracranial EEG recorded while participants performed a paired associates cued recall task. The specific question was whether there patterns of functional connectivity related to encoding and retrieval of specific items.

While analysis of functional connectivity has greatly expanded our understanding of the neural correlates of memory encoding and retrieval, it has done so on the average level, due to either the slow time course of the data analyzed (e.g., fMRI) or the general approach of aggregating over many events. To address this shortcoming, the authors developed a novel approach that allowed for assessing the dynamic changes in functional connectivity arising from individual encoding and retrieval events. A critical aspect of their findings is that the dynamic changes in functional connectivity they observed are not due to general effective memory states, but actually track item-specific encoding and retrieval processes.

This manuscript is an important contribution to the field, demonstrated through a creative and sophisticated analysis of a neural activity collected while participants performed a canonical recall memory paradigm. The methods, while complicated, are solid and explained clearly. Below I list some minor comments that I hope assist the authors as they revise their manuscript.

We thank the Reviewer for their kind words and positive feedback.

- p5: It was clearly stated that no event locking was used for the initial WSCC analysis, but was it conducted with time periods that included the task? Based on Figure S1 I assume these blocks came from task periods, but can you be more specific what is meant by "random recording time periods"? Note, I see this in the methods later, but it may be good to further discuss why you picked random chunks of data. On a related note, do you get the same connectivity patterns (and time delays) if you perform this initial WSCC analysis on data from a completely different time/task?

We thank the Reviewer for their comments and agree that this is an important point to clarify. A similar point was made by Reviewer #1, and we briefly summarize our above response here:

These random time periods are sampled from entire iEEG recording sessions, which include both task and non-task periods. These non-task periods include, for example, times when participants are speaking with hospital staff, taking a phone break, taking a bathroom break, etc. We adopted this approach because previous work in our lab has demonstrated that these network connections and their time delays remain stable across minutes, hours, and even days, regardless of which specific behavior an individual is exhibiting during that time (Chapeton 2017, Xie 2023). Additionally, almost all participants in our study here completed 2 separate experimental sessions of this task (17 of 20 participants). The timing between sessions, often hours or days apart, is now shown in Supplementary Table S3. We only selected electrode pairs that exhibited significant connectivity in both sessions, ensuring that the observed connectivity in these electrode pairs was not specific to only one recording session.

Furthermore, we observe that an example electrode pair of one subject has a coupling profile that is reliable and stable across three different time sampling methods: across the recording session, only from encoding trials, and only during successful encoding trials (figure shown again below for convenience). This example suggests that the coupling profiles we identify are consistent, regardless of whether we use time blocks distributed throughout the entire task or only during encoding task time. We have now clarified all these points in our revised Methods.

Finally, it is also important to clarify that our method does not necessarily select for pairs that show only elevated connectivity in response to the stimulus. Around half of the pairs we have identified with overall strong connectivity when assessed using random time points throughout the experimental session in fact exhibit a decrease in connectivity in response to stimulus during the actual memory task itself. Moreover, even if a pair is identified with increases in coupling during both encoding and retrieval task time periods, these overall increases in connectivity are not relevant for our analysis. Instead, once we select a connected pair, we are interested in how that pair's coupling changes over encoding versus retrieval at its specific maximum time lag. As shown above in our illustrated example, the max time lag for a given pair remains stable regardless whether we sample our time blocks for cross correlation analysis from across the entire recording session, encoding time period, or successful encoding time periods. However, from moment to moment, that connectivity or coupling between that electrode pair will exhibit dynamic changes, and our primary interest here is in how those dynamic changes may be reinstated during successful memory retrieval and specific to the item or association being encoded into and retrieved from memory.

- p5: I realize you refer to the methods here, but might it be possible to give a one-sentence description of the "common thresholding heuristic" used to select electrode pairs? Or is the description for the rest of this paragraph summarizing that heuristic? As written it seems like there was a threshold applied and then the pairs were further reduced.

We thank the Reviewer for this helpful suggestion. We have now added a one-sentence description of the "common thresholding heuristic" in our Methods. Our description is also shown here in red:

"To select electrode pairs that exhibit strong time-locked correlations, we used a thresholding method based on the distributions of maximum coupling values and peak sharpness (coincidence index) values across pairs' coupling profiles (see Figure 1d; Figure 1e; see Methods)."

We have also added a reference to Figure 1d, since the figure shows a schematic of how the coincidence index values are calculated (see below).

- p7: The shuffling procedure to demonstrate the item-specific reinstatement of functional connectivity patterns is extremely important and one of the most interesting results presented. Did the same hold for the spectral analysis?

We thank the Reviewer for raising this question. We have previously shown that yes, indeed, item-specific reinstatement is also apparent when evaluating patterns of spectral power analyses (Yaffe 2014). This previous study was conducted using the same task and data from overlapping participants in our current study. We have recreated the results of that analysis from that previous study below:

when we shuffled trial labels. We first shuffled the labels of all retrieval periods and calculated reinstatement between the true correct encoding periods and the shuffled retrieval periods. We found that the average reinstatement in the tROI originally observed with our data were significantly greater than reinstatement in this shuffled distribution [$t_{(29)} = 8.50$, $P < 10^{-4}$, paired, two-tailed; Fig. 2 C and D, Shuffle All]. We next shuffled the labels of only the correct retrieval periods and calculated reinstatement between the true correct encoding periods and the shuffled correct retrieval periods. If reinstatement reflects a general encoding and retrieval process, then reinstatement calculated using shuffled correct retrieval periods should be identical to that observed using the original correct trials. Instead, we found that mean reinstatement in the tROI was significantly less when we used the shuffled compared with the true correct trials [$t_{(29)} = 6.58$, $P < 10^{-4}$, paired, two-tailed; Fig. 2 C and D, Shuffle All Correct; see also Fig. S4]. When we restricted shuffling to only swap retrieval periods from adjacent correct trials, we also found that mean reinstatement in the tROI during correct trials was significantly greater than that calculated using the shuffled adjacent correct trials [$t_{(29)} = 4.81$, $P = 0.00004$, paired, two-tailed; Fig. 2 C and D, Shuffle Adjacent Correct].

To clarify this point, we have now added this citation in our Results when we discuss the item-specific reinstatement of coupling patterns.

- p8, l221: Is the claim that there is not a significant difference in reinstatement power between the high vs. low coupling electrodes resting on the $p=0.053$? Meaning that it's just above the standard 0.05 threshold? If so, that's a quite weak claim. If not, can you explain your test in more detail? Also, Figure S8 has the caption Figure S6, such that there are two S6 figures. That said, it does not seem like Figure S6b addresses the claim. I am, however, convinced by the entire pattern of results for the split-half analyses that the spectral and coupling reinstatement is dissociable.

We thank the Reviewer for inviting us to clarify this point. Our primary claim here is not that power reinstatement between high vs. low coupling electrodes is insignificant; instead, our emphasis is on the interaction effect between our two different comparisons: 1) coupling reinstatement between high vs. low coupling electrodes and 2) power reinstatement between high vs. low coupling electrodes. It is the difference between these two comparisons that we hope to emphasize. The same holds true for our parallel analysis looking at coupling vs power reinstatement for high vs. low power electrodes.

We have made changes in our Results to help clarify these claims. We have also made changes to Figure 4b and 4d by adding significance bars for the interaction effect between reinstatement analyses and clarifying the interaction effect in the Figure caption. We show these changes here:

Results:

“However, splitting electrodes in the same manner does not lead to a significant similar change in the reinstatement of power within the tROI ($t(19) = 2.06$, $p = 0.053$, Cohen’s $d = 0.46$; Figure 4b), see tROI of spectral power analysis in Supplementary Figure S9b). Instead, splitting the electrodes based on their relative contributions to the reinstatement of coupling has led to a significantly greater change in the reinstatement of coupling as compared with that of spectral power (interaction between electrode group and reinstatement measure: $F(1, 19) = 37.52$, $p = 0.000069$, $\eta^2 = 0.66$) (Figure 4b).”

“However, this split does not lead to significant similar changes in the reinstatement of coupling ($t(19) = 1.45$, $p = 0.16$, Cohen’s $d = 0.32$; Figure 4d, see tROI of coupling analysis in Figure 3c). Instead, splitting the electrodes based on their relative contributions to the reinstatement of spectral power only leads to a significantly greater change in the reinstatement of spectral power but not in the reinstatement of coupling (interaction between electrode group and reinstatement measure: $F(1, 19) = 18.21$, $p = 0.00042$, $\eta^2 = 0.49$) (Figure 4d).”

Figure 4 caption:

Fig 4b) “...When splitting electrodes based on their contributions to the reinstatement of coupling, there is a significantly greater change in the reinstatement of coupling as compared with that of spectral power.” and Fig 4d) “...When splitting electrodes based on their contributions to the reinstatement of power, there is a significantly greater change in the reinstatement of spectral power but not in the reinstatement of coupling.”

We also apologize for the typo in referencing the appropriate Supplementary Figure. We have fixed this in the Results and now refer to the correct figures for the tROIs for coupling and power reinstatement, also shown above.

- p11, l282: Drawing RTs from correct trials to fill in the "pass" trials seems like you're making up data. Why do this at all? I read awhile later in the methods that this is to specify the range for non-responses. It's likely worth mentioning this here.

We thank the Reviewer for raising this point. Trials without a vocalization do not have a response time (RT) by nature. In order to analyze neural data across all available trials in order to make a comparison between correct and incorrect memory retrieval, we randomly assign pass trials a RT from the distribution of actual RTs across trials with vocalization for that participant. We have made relevant changes in Methods for clarification, also shown here:

“During pass trials where no vocalization was present, we assigned a response time by randomly drawing from the distribution of correct response times during that experimental session **in order to constrain the range of possible response times and align all trials by time for subsequent analyses**”

- p11, l305: How do you specify the location of the bipolar referenced electrodes? Do you split the difference between the pair?

Yes, this is correct. To specify the location of bipolar referenced electrodes, we take the midpoint between the locations of two adjacent recorded electrodes. We have clarified this in Methods, also shown here:

“We re-referenced the resulting signals using bipolar referencing based on the immediate adjacent electrode contacts to mitigate any effects of volume conduction or any biases introduced by the system hardware reference. **The location of bipolar-referenced signals were defined by the midpoint between adjacent electrode contacts.** Henceforth, we refer to these bipolar-referenced signals as electrodes.

- f3: There is a subfigure e in the caption that does not exist in the figure.

We apologize for the typo and have now corrected the caption in Figure 3.

Reviewer #2 (Remarks on code availability):

Although I did not download the data, I only saw a data download at that link, not a separate download for code. It could be that there is code provided with the data.

Yes, our download link includes both code and data.

Reviewer #3 (Remarks to the Author):

In the paper titled 'Dynamic Patterns of Functional Connectivity in Human Cortical Networks Are Specific to Individual Memory Formation,' the authors explored the role of functionally connected electrode pairs in memory formation. They observed a higher connection strength between electrode pairs in correct trials compared to incorrect ones during encoding. Additionally, they found the reinstatement of the functional connectivity pattern during the retrieval session for word-association. Furthermore, they dissociated the reinstatement of functional connectivity pattern and power pattern. While I found the results intriguing, there are a few aspects that the authors could clarify:

We appreciate the Reviewer's comments and hope to provide some clarification with our responses below.

1. To eliminate potential influences from vocal preparation on encoding-retrieval similarity differences between correct and incorrect trials. It might be beneficial to segregate the incorrect condition into wrongly responded trials and non-response trials.

We thank the Reviewer for this suggestion. We agree that potential influences from vocal preparation are important to address. We have conducted a new analysis recomputing our reinstatement analysis for correct trials vs. intrusions (incorrect vocalized responses) and correct trials vs. passes (trials where participants vocalized "pass" or did not respond). All new analyses here are added to a new Supplementary Figure S6.

Given the limited amount of intrusion trials for each participant (see Supplementary Table S2), we only included participants with at least 10 intrusion trials in our correct trials vs. intrusions analysis. This resulted in 12 participants (out of original 20) which we included for our analysis.

We were able to replicate our original findings when comparing correct trials with pass trials across all 20 participants (Supplementary Figure S6a). That is, correct trials show significantly greater similarity patterns of coupling between encoding and retrieval than pass trials. We did not find significant reinstatement for correct trials compared to intrusions for the 12 participants with at least 10 intrusion trials, although the mean level of reinstatement appears to be larger for correct trials (Supplementary Figure S6b):

We believe the absence of a significant difference between correct trials and intrusions may be related to two points. First, even participants with greater than 10 intrusion trials had very few intrusion trial counts in general and also compared to correct trial counts (see Supplementary Table S2). These low trial counts could contribute to the noise of the data. Second, we believe this point may be related to a point raised by Reviewer 1 in which we examine the semantic similarity between items. As seen in previous work (Jang 2017), the similarity between neural signals during correct trials and intrusions may be partially attributed to the semantic nature of the intrusions. For intrusions, we now show that the vocalized word (intrusion) is more semantically similar to the cue word than the expected word, which may explain why we still see some reinstatement in these cases (Supplementary Figure S6d). These findings suggest that when participants make an intrusion, they are activating many of the same patterns of connectivity that are active when studying the original word pair. We have referenced our new analysis the revised Results and have replicated our findings presented in the new Supplementary Figure S6 below:

2. Could the author describe the behavioural performance clearly?

We agree with the Reviewer that providing these additional details would be helpful. In our revised Results, we now report the percentage of correct trials and the median reaction time. In Supplementary Table S2, we also provide additional information for each participant regarding the number of intrusion and pass trials. We have also added a reference to Supplementary Table S2 in Results.

3. Could the authors provide information on the distribution of T_{max} ? Does the spatial distance between connected electrode pairs and non-connected pairs differ? How does the autocorrelation attenuate with time within each electrode? Will the auto-correlation affect the cross-correlation?

We thank the Reviewer for their questions and suggestions. The first two points are addressed in a new Supplementary Figure S3 and referenced accordingly in the Results. We have also provided the distribution of T_{max} across all electrode pairs in Supplementary Figure S3a as shown below.

We also now introduce a new analysis, based upon the Reviewer's suggestion, to examine the spatial distances between connected and non-connected electrode pairs. We found that electrodes of unselected pairs were significantly further apart than electrodes of selected pairs (Supplementary Figure S3b). We believe these findings are consistent with prior studies that have demonstrated that most connections are relatively local. There are indeed strong connections across larger distances of cortex, but these long-range connections tend to be less frequent.

The Reviewer also raises an interesting question about the auto-correlation. It is possible that if two electrodes each have relatively wide auto-correlations, then their cross-correlation will also be relatively wide. This is because in our case, the samples contributing to the cross-correlation are not truly independent by virtue of the inherent auto-correlation in each signal. Even in these cases, however, there are two important points to note. First, the criteria we use to select an electrode pair for inclusion rely both upon the magnitude and relative width of the cross-correlation. The magnitude of the cross-correlation between two signals that exhibit strong auto-correlations may be higher. However, we do not simply use the magnitude or the 95% confidence interval to determine whether that pair is eligible for inclusion. Instead, we specifically compare the

normalized maximum correlation across all electrode pairs to identify significant magnitudes, and a coincidence index to identify cross-correlations with sharp peaks. If the cross-correlation between two auto-correlated signals does not satisfy these criteria, we would not include that electrode pair in our analyses. Second, and perhaps more importantly, we use the cross-correlation to identify eligible electrode pairs. For each identified pair, we are then able to identify a preferred time lag, and all subsequent analyses are performed on the dynamic changes in cross-correlation in that electrode pair only at that specific time lag. Thus, whether the cross-correlation is narrow or wide does not impact the subsequent analyses, as long as the cross-correlation is strong and narrow enough to satisfy the inclusion criteria. Instead, our primary interests is on the dynamic changes in cross-correlation at this specific time lag during memory encoding and retrieval for each trial.

4. Typo here? 'Across participants, the magnitude of changes in coupling strength increases from baseline during the encoding period and peaks around -1000 to -500 ms before response vocalization.'

We thank the Reviewer for bringing up this point for clarification. We have now fixed all of our notation to the format of “-1000 to -500 ms **relative to** response vocalization”, as shown below. We have also clarified in our Results that time of vocalization is denoted by retrieval time of 0. Thus, negative time values indicate time points before the moment of response vocalization.

*“Across participants, the magnitude of changes in coupling strength increases from baseline during the encoding period and peaks around -1000 to -500 ms ~~before~~ **relative to** response vocalization, **which is referred to as retrieval time = 0 s.**”*

*“In individual participants, we observed that some electrode pairs exhibit dynamic increases in coupling ~~relative~~ **compared** to baseline periods (-1000 ms to -500 ms **relative to response vocalization**) following stimulus onset during the encoding period, while other pairs exhibit dynamic decreases relative to baseline periods (Figure 2b; see Methods).”*

*“...and -550±170 ms ~~before~~ **relative to** vocalization”*

*“This tROI peaks around 1,050 ± 340 ms after study onset and -1,000 ± 330 ms ~~before~~ **relative to** vocalization, replicating the previous findings”*

5. Exploring how the reinstatement of connectivity pattern and power pattern, when dissociated, relates differently to behavioral performance is crucial. Additionally, elucidating the spatial distributions of the involved electrodes, such as the differing brain regions, would be valuable.

We thank the Reviewer for this question. To clarify, the behavioral data that we use for both the analysis of power reinstatement and the analysis of coupling reinstatement is the same. When examining the dissociation of power versus connectivity and how they relate to behavioral performance, we analyze the same set of trials and behavioral task sessions, only altering how we analyze the neural data. In our current study, we only have two major behavioral categories of interest: correct responses and incorrect responses. We believe that correct recall requires both the reinstatement of spectral power and patterns of coupling, and that these mechanisms are not redundant with each other.

The spatial distributions of involved electrodes can be found in Supplementary Figure S10.

Supplementary Figure 10

Supplementary Figure 10. Electrode contributions to coupling and spectral power reinstatement. Spatial distribution of electrodes across participants and their relative contributions (percentile) to the reinstatement of spectral power and to the reinstatement of dynamic connectivity. We do not observe any regional localization of brain areas that are more or less important for the reinstatement of spectral power or for the reinstatement of dynamic connectivity. Reinstatement contribution percentiles were calculated using a leave-one-out approach (see Methods).

We have also clarified this in the Results as shown below.

*“We did not observe specific regional localization of brain areas that contribute more towards the reinstatement of **spectral power** or dynamic connectivity (see Supplementary Figure 10 for **spatial electrode distributions**).*

6. Could the authors provide information on the distribution of functional connectivity between electrodes contributing to power pattern reinstatement?

We thank the Reviewer for this suggestion. We have now included the distribution of Wmax values for all selected electrode pairs in the new Supplementary Figure S3. To further illustrate the distribution of Wmax between electrodes contributing to power and connectivity reinstatement, we calculated average Wmax across electrode pairs in which each of two electrodes fell into the categories of “high contribution to power reinstatement”, “low contribution to power reinstatement”, “high contribution to connectivity reinstatement”, and “low contribution to connectivity reinstatement”. We compared average Wmax values among high vs. low contribution to power reinstatement electrodes, high vs. low contribution to connectivity reinstatement electrodes, and high contribution to connectivity vs. high contribution to power reinstatement electrodes, shown below. We did not find a significant difference in average Wmax values between these groups, suggesting that the Wmax values of electrode pairs are independent of how they contribute to reinstatement of coupling and spectral power.

Thank you again for considering our manuscript. We look forward to your reply and to the reviews of our manuscript.

Sincerely,

Kareem A. Zaghloul, MD, PhD
 Surgical Neurology Branch, NINDS
 National Institutes of Health
 Building 10, Room 3D20
 10 Center Drive
 Bethesda, MD 20892-1414
 O: (301) 594-8114
 F: (301) 402-0380
 kareem.zaghloul@nih.gov

Reviewer #1 (Remarks to the Author):

I apologize for the lack of clarity in my previous comments. It is a little different to say that lag=0 is removed due to volume conduction versus reflecting a common input source. The current language is mixing these ideas. If the latter is the concern, then there should be a range of time values that are within a putative synaptic delay that are excluded, not just lag=0. Moreover, the methods used by the authors do not really establish that there are direct synaptic connections between different network regions anyway, and in that sense I think many regions will reflect co-activation in the setting of a common input region, even at other lag values. If the concern is volume conduction, then connections over longer distances should not be eliminated even at lag=0. My suggestion is that the manuscript would benefit from demonstrating how results differ if they accept lag=0 connections at distances long enough to avoid volume conduction, as I don't think there is good reason to exclude lag=0 otherwise.

In regards to my second comment, I apologize again for the lack of clarity. I understand that participants must respond within 5 seconds, but I was asking if they are allowed to respond at any point within the 5 seconds or if they must wait to verbalize the associate until a specific time lag has passed. It sounds like it is the former. In that case, there could be a difference in the dynamics of trials in which they answer fast versus slow. I think the manuscript would benefit from understanding how the response times modulate the connectivity dynamics they are observing. My comment is pretty similar to that of Reviewer 2. If the subjects responds quickly, there may be a mix of cue—related processing and item recollection activity (if they respond within 1.5 sec, then a lag of -2 sec from vocalization is harder to interpret) ALternativley, using the response time as a regressor would identify modulation that is not sensitive to the RT.

The authors did a good job responding to my third comment. I suppose the relevance of the comments depends on the percentage of the random time blocks occupied by encoding/retrieval segments in general – if it is particularly high, then the circularity concern would apply. The individual example data they include suggest that the connectivity method is not sensitive to exclusion of encoding/retrieval segments, which helps support the use of this method. The manuscript would benefit from showing that the headline results, ie Figure 3, exhibit the same relative insensitivity since it is a core result.

The analysis the authors include related to intrusions is interesting. It is reassuring to see this expected behavioral pattern, as it is a core prediction of context models (along with primacy items becoming intrusions in other lists). I don't really understand why a semantic similarity estimate between a pair of words and another pair of words can't be generated as a composite from word2vec adjacencies, or any number of other ways really. Perhaps I am missing something. Either way, I think my comment missed the mark somewhat. There are better ways to identify activity linked with individual items – authors could consider the methods for concept neurons based on vector characteristics relative to a distribution of all other items as described by Kolibius <https://www.nature.com/articles/s41562-023-01706-6>. I think it would make the item—level results more convincing, but I wouldn't demand that the authors do this since my original suggestion was not very good. I don't think it is especially surprising that an item pair is more similar to itself than other items, so further specification of this result is probably overkill.

I think the updated methods probably account for the output serial presentation. If the order of

items at test is random, it probably doesn't matter unless the authors were trying to identify contiguity effects.

Reviewer #1 (Remarks on code availability):

Seems ok. Readme is sufficient

Reviewer #2 (Remarks to the Author):

The authors have done a commendable job addressing the concerns of all the reviewers. I have no further concerns or suggestions.

Reviewer #2 (Remarks on code availability):

The authors claim responded that their data download also contained the code. This is not obvious from the webpage listed above, which only has a link for Data and not a separate link for Code, which they have on other manuscripts. Not knowing how large the file would be, I did not test the Data download.

June 21, 2024

Thank you again for the opportunity to revise our manuscript, “Dynamic patterns of functional connectivity in human cortical networks are specific to individual memory formation” for publication in *Nature Communications*. We thank the reviewers again for their insightful feedback that we find very helpful and that continue to improve our manuscript. Here we have provided some new analyses and revised our manuscript based on their comments, which we feel have further strengthened our findings and conclusions.

The reviewer comments are reproduced in bold italic font, and we have addressed each point below the corresponding comment. We have provided a version of the manuscript highlighting all the relevant changes using red font as well as a clean pdf version of the revised manuscript.

Reviewer #1 (Remarks to the Author):

I apologize for the lack of clarity in my previous comments. It is a little different to say that lag=0 is removed due to volume conduction versus reflecting a common input source. The current language is mixing these ideas. If the latter is the concern, then there should be a range of time values that are within a putative synaptic delay that are excluded, not just lag=0. Moreover, the methods used by the authors do not really establish that there are direct synaptic connections between different network regions anyway, and in that sense I think many regions will reflect co-activation in the setting of a common input region, even at other lag values. If the concern is volume conduction, then connections over longer distances should not be eliminated even at lag=0. My suggestion is that the manuscript would benefit from demonstrating how results differ if they accept lag=0 connections at distances long enough to avoid volume conduction, as I don't think there is good reason to exclude lag=0 otherwise.

We thank the Reviewer raising this point and apologize for any confusion from our response in the previous response letter. It is important to acknowledge that the concerns of volume conduction versus common input source are separate. The concern of common inputs is a valid one in any functional connectivity analysis, and as the Reviewer correctly identifies, common inputs could give rise to apparent functional connections with non-zero time delays. We have previously demonstrated that for the sites that we are recording from, most effective connections between electrode pairs do not arise from common input or transitivity from the other electrodes (Chapeton 2017, Supplementary), however, we can never rule out the effects of common inputs arising from sources that we are not recording from. As the Review also correctly infers, our main concern and goal for removing connections with lag = 0 was to eliminate the effects of volume conduction.

The Reviewer brings up an interesting point about accepting lag = 0 connections at distances long enough to avoid volume conduction. However, there is not a standard distance we are aware of at which volume conduction can be disregarded, so we opted to do a new analysis where we include all lag = 0 connections. In this manner, we may be including some spurious connections, but we then also include true connections that may have been missed with our original analysis excluding lag = 0 connections. Averaged across all participants, we show our reinstatement of coupling results here, also in Supplementary Figure S14. Our data suggests that our results appear similar with or without lag = 0 connections.

These data therefore suggest that our primary results are not confounded by the elimination of connections with lag=0.

We have also clarified this point here in the text: “We also removed all electrode pairs where the W_{max} occurred with zero time delay, $\tau_{max} = 0$, since these functional connections can arise spuriously, for example, due to volume conduction (see Supplementary Figure S14 for results after inclusion of these electrode pairs).” To summarize, eliminating lag = 0 connections in our original analysis may exclude some real interactions, but ensures that spurious coupling does not confound our results.

In regards to my second comment, I apologize again for the lack of clarity. I understand that participants must respond within 5 seconds, but I was asking if they are allowed to respond at any point within the 5 seconds or if they must wait to verbalize the associate until a specific time lag has passed. It sounds like it is the former. In that case, there could be a difference in the dynamics of trials in which they answer fast versus slow. I think the manuscript would benefit from understanding how the response times modulate the connectivity dynamics they are observing. My comment is pretty similar to that of Reviewer 2. If the subjects responds quickly, there may be a mix of cue—related processing and item recollection activity (if they respond within 1.5 sec, then a lag of -2 sec from vocalization is harder to interpret) Alternatively, using the response time as a regressor would identify modulation that is not sensitive to the RT.

We thank the Reviewer for bringing up this point and for allowing us to clarify. Participants are allowed to respond at any point within the 5 seconds (see Methods: “Participants can vocalize their response any time during the retrieval period after cue presentation.”)

The average median reaction times (RTs) across participants are stated in the Results, and the median reaction times per participant can be found in Supplementary Table S2 (distribution shown here as well). The average median RT is 2.1 seconds +/- 0.1 seconds, and the mean reinstatement effect peaks around -0.55 seconds before vocalization (see Results and Figure 3b-c), which suggests that, on average, the peak effect is seen right before response vocalization and not, for example, prior to the presentation of the cue. We have reproduced those main results regarding the distribution of response times and the effect of reinstatement from our original submission here for clarity:

We appreciate the Reviewer’s suggestion about investigating how response times modulate connectivity dynamics. As the Reviewer suggested, we introduced a new analysis to address the relationship between response times and connectivity dynamics. We recomputed the main connectivity reinstatement effect (Figure 3b-c) between “fast” vs. “slow” correct responses. For each participant, using that participant’s median reaction time, we split their correct responses into “fast” responses (RT < median RT) or “slow” responses (RT > median RT). Averaging across participants, we find that the reinstatement effects appear similar for both “fast” vs. “slow” correct responses. Thus, this new analysis suggests that speed of response times should not affect our results. We present this new analysis in a new supplementary figure, Supplementary Figure S7.

The authors did a good job responding to my third comment. I suppose the relevance of the comments depends on the percentage of the random time blocks occupied by encoding/retrieval segments in general – if it is particularly high, then the circularity concern would apply. The individual example data they include suggest that the connectivity method is not sensitive to exclusion of encoding/retrieval segments, which helps support the use of

this method. The manuscript would benefit from showing that the headline results, ie Figure 3, exhibit the same relative insensitivity since it is a core result.

We thank the Reviewer for their positive feedback and follow-up comments concerning this point. As mentioned by the Reviewer, we calculated the percentage of the random time blocks occupied by encoding/retrieval segments to contextualize the relevance of these concerns. We found that across sessions, the average percentage of timepoints from encoding/retrieval that overlaps with the randomly selected blocks is 30.6%, suggesting that about 70% of the data used for identifying functional connectivity are not affected by the task. We notice that even when using data overlapping or non-overlapping with task periods, the identified connected electrode pairs show highly similar cross-correlation profiles with highly consistent lag, not only in the current data, but also in prior data using this method (Xie 2023).

Furthermore, as the Reviewer mentions, the individual example data from our previous revision suggests that the connectivity method is not sensitive to encoding/retrieval segments. However, as suggested by the Reviewer, we computed the reinstatement analysis (i.e., our “headline results” in Figure 3) for one participant when sampling time blocks only from encoding trials. Our results, shown here, suggest that these results exhibit the same relative insensitivity to sampling from trial time periods. We have now included this analysis, as well as our findings from the previous response letter, in our manuscript as Supplementary Figure S13.

These results are in line with previous results which show that these network connections and their time delays remain stable across minutes, hours, and days, regardless of whether an individual is sleeping, eating, talking, doing a task, doing a different task, or resting (Chapeton 2017, Xie 2023). Together, we conclude that both the connectivity profiles and our main reinstatement effect are consistent whether we use time blocks distributed throughout the entire task or only times during the encoding trials.

The analysis the authors include related to intrusions is interesting. It is reassuring to see this expected behavioral pattern, as it is a core prediction of context models (along with primacy items becoming intrusions in other lists). I don't really understand why a semantic similarity estimate between a pair of words and another pair of words can't be generated as a composite from word2vec adjacencies, or any number of other ways really. Perhaps I am missing something. Either way, I think my comment missed the mark somewhat. There are better ways to identify activity linked with individual items – authors could consider the methods for concept neurons based on vector characteristics relative to a distribution of all other items as described by Kolibius <https://www.nature.com/articles/s41562-023-01706-6>. I think it would make the item—level results more convincing, but I wouldn't demand that the authors do this since my original suggestion was not very good. I don't think it is especially surprising that an item pair is more similar to itself than other items, so further specification of this result is probably overkill. I think the updated methods probably account for the output serial presentation. If the order of items at test is random, it probably doesn't matter unless the authors were trying to identify contiguity effects.

We appreciate the Reviewer's positive comments towards our added analyses. We also agree that the paper that the Reviewer references is an excellent paper and we appreciate the Reviewer bringing it to light. In response to the Reviewer's question about measuring semantic similarity between different word pairs, we still think it is quite challenging to represent the semantic content of a word pair since each pair is made up of two separate words with different semantic values. Generating a composite, for example, might not completely capture the semantic content of the pair as a whole. There are multiple ways that one could construct a conjunction between two words, and each construction would involve some assumptions. We agree that exploring the relation between the similarity between two pairs of words and the similarity of neural connectivity would be a very interesting point, and we appreciate the Reviewer bringing this up. However, given its complexity, we think the most we can assert here is that the patterns of connectivity are specific to the individual trial, and that exploring the relation with semantic similarity may be out of scope for our current manuscript. We do think it is important in future research to test how semantic similarity between word pairs may be linked to neural features. Importantly, our current findings still hold regardless of the semantic values of the randomly selected word pairs.

We thank all the Reviewers for their time, insights, and valuable feedback. We appreciate your consideration of our manuscript and look forward to your reply.

Sincerely,

Kareem A. Zaghloul, MD, PhD
Surgical Neurology Branch, NINDS
National Institutes of Health
Building 10, Room 3D20
10 Center Drive
Bethesda, MD 20892-1414
O: (301) 594-8114
F: (301) 402-0380
kareem.zaghloul@nih.gov

References

Chapeton, J. I., Inati, S. K. & Zaghoul, K. A. Stable functional networks exhibit consistent timing in the human brain. *Brain* 140, 628–640 (2017).

Xie, W. et al. The medial temporal lobe supports the quality of visual short-term memory representation. *Nature Human Behaviour* 1–15 (2023).

Reviewer #1 (Remarks to the Author):

I appreciate the diligence in responding to the comments. The authors should be congratulated on a timely and interesting paper.